# Viral potential to modulate microbial methane metabolism varies by habitat

Zhi-Ping Zhong [1,2,3], Jingjie Du [2,10], Stephan Köstlbacher [4,5,11], Petra Pjevac [4,6], Sandi Orlić [7,8] & Matthew B. Sullivan [1,2,3,9]

Methane is a potent greenhouse gas contributing to global warming. Micro-organisms largely drive the biogeochemical cycling of methane, yet little is known about viral contributions to methane metabolism (MM). We analyzed 982 publicly available metagenomes from host-associated and environmental habitats containing microbial MM genes, expanding the known MM auxiliary metabolic genes (AMGs) from three to 24, including seven genes exclusive to MM pathways. These AMGs are recovered on 911 viral contigs predicted to infect 14 prokaryotic phyla including Halobacteriota, Methanobacteriota, and Thermoproteota. Of those 24, most were encoded by viruses from rumen (16/24), with substantially fewer by viruses from environmental habitats (0–7/24). To search for additional MM AMGs from an environmental habitat, we generate metagenomes from methane-rich sediments in Vrana Lake, Croatia. Therein, we find diverse viral communities, with most viruses predicted to infect methanogens and methanotrophs and some encoding 13 AMGs that can modulate host metabolisms. However, none of these AMGs directly participate in MM pathways. Together these findings suggest that the extent to which viruses use AMGs to modulate host metabolic processes (e.g., MM) varies depending on the ecological properties of the habitat in which they dwell and is not always predictable by habitat biogeochemical properties.

Earth is currently warming at an unprecedented speed over at least the last 2000 years[1], partly owing to the increased concentration of greenhouse gases in the atmosphere[2]. Methane ($CH_4$) is ranked second after carbon dioxide ($CO_2$) in terms of the overall contribution to atmospheric warming and accounts for ~20% of the greenhouse gas-driven warming[3–5]. Approximately 50% of global methane emissions originate from aquatic ecosystems, of which freshwater lakes contribute up to 53%[6]. Wetlands are another important natural source of

methane, whereas non-water-logged terrestrial ecosystems generally function as methane sinks[7]. Methane cycling is largely driven by microbes, with microbial methanogenesis (all mediated by archaea) producing ~69% of the total methane released to the atmosphere[8]. Among anthropogenic sources, about 30% of methane production is microbially mediated, almost all of which derives from ruminant live-stock farming[7]. Understanding how cellular microbes and the viruses that infect them might impact methane metabolism (MM) across

[1]Byrd Polar and Climate Research Center, Ohio State University, Columbus, OH, USA. [2]Department of Microbiology, Ohio State University, Columbus, OH, USA. [3]Center of Microbiome Science, Ohio State University, Columbus, OH, USA. [4]Division of Microbial Ecology, Department of Microbiology and Ecosystem Science, Centre for Microbiology and Environmental Systems Science, University of Vienna, Vienna, Austria. [5]Doctoral School in Microbiology and Environmental Science, University of Vienna, Vienna, Austria. [6]Joint Microbiome Facility of the Medical University of Vienna and the University of Vienna, Vienna, Austria. [7]Division of Materials Chemistry, Ruđer Bošković Institute, Zagreb, Croatia. [8]Center of Excellence for Science and Technology-Integration of Mediterranean Region, Zagreb, Croatia. [9]Department of Civil, Environmental and Geodetic Engineering, Ohio State University, Columbus, OH, USA. [10]Present address: Division of Nutritional Science, Cornell University, Ithaca, NY, USA. [11]Present address: Laboratory of Microbiology, Wageningen University and Research, Wageningen, the Netherlands. ✉e-mail: sorlic@irb.hr; sullivan.948@osu.edu

various habitats is therefore crucial to inform efforts to mitigate microbially driven methane emission and climate warming.

The impact of viruses on MM has only recently begun to be investigated[9–11]. Viruses are found ubiquitously in the environment and have important roles in ecological, biogeochemical, and evolutionary processes through cells lysis, horizontal gene transfer, and modulation of host metabolism (including carbon, sulfur, and nitrogen metabolism)[12–15]. Recently, some viruses have been found to encode *pmoC* and *cofF* as auxiliary metabolic genes (AMGs), with the potential to supplement the aerobic oxidation of methane by their bacterial host in freshwater lakes[9]. In addition, putative *cofF* and *fae* genes were found in viruses from deep-sea hydrothermal vents[11]. Notably, MM AMGs were recently also found in a novel group of extrachromosomal elements called "Borgs", which have been shown to encode *mcr* genes phylogenetically related to those of anaerobic methanotrophic archaea (ANME) in the genus *Methanoperedens*[16]. However, beyond these initial observations, little is known about other MM genes encoded by viruses or other extrachromosomal elements, or how they influence methane production.

Here, we sought to explore the potential effects of viruses on MM in habitats with microbially-derived MM. First, we analyzed 982 publicly available metagenomes from a range of environments, which are known from literature to potentially host methane-cycling microbial communities, and in which we were able to confirm the presence of microbial MM genes, to identify virus-encoded AMGs that could be involved in the MM pathway (MMP), including those participating in methane production (i.e., methanogenesis by archaea) and oxidation (either by aerobic methanotrophic bacteria or anaerobic methanotrophic archaea). We then generated an additional 11 metagenomes using lake sediments, in which methane emission has been detected, from Croatia's largest freshwater lake (Vrana Lake)[17] to sample bacterial/archaeal viruses and investigate their potential impacts on MM during infection.

## Results

### Some viruses encode genes that could modulate microbial methane metabolism

To discover new MM AMGs, 982 publicly available metagenomes from 15 environments (Supplementary Data 1; including rumen, marine water, marine sediment, lake water, lake sediment, river estuary sediment, wetland sediment, and permafrost active layers, among others), were analyzed for microbial genes involved in MMP and viral genomes encoding MM AMGs. The assembled contigs excluding viral contigs, were used to identify microbial genes involved in MMP (based on their KEGG and PFAM annotations and the KEGG MM pathway modules[18]) and each of the environments contained 138–183 distinct microbial genes involved in MMP (in total 184 distinct genes from all environments after dereplication; Supplementary Data 2 & 3). Viral genomes were also identified from the assembled contigs of these metagenomes, using a combination of three tools: VirSorter[19], DeepVirFinder[20], and MARVEL[21] (see Methods). In the identified viral genomes, we predicted and annotated viral genes and screened for putative virus-encoded MM AMGs using VIBRANT[22] and manual curation. Particularly, MM AMGs were extracted based on their KEGG annotations and the MM pathway modules[18]. After rigorous inspection (see Methods), 911 viral contigs were identified to contain MM AMGs (Supplementary Data 4), resulting in the discovery of 24 distinct AMGs that potentially participate in 25 metabolic reactions in the MMP (Supplementary Data 5, Figs. S1 and S2). These 911 viral contigs originated from ~32% (316 of 982) of the here analyzed metagenomes (Supplementary Data 3). We compared the 911 viral contigs to the viral genomes/contigs from the NCBI RefSeq database (cultivable viral genomes) and IMG/VR database (uncultivated viral genomes from metagenomics)[23] using a genome-based network approach (see Methods)[24,25]. About 34% of these viruses (308 of 911) could be assigned to taxonomy and all

belonged to the class Caudoviricetes of the phylum Uroviricota, except one that belonged to an unclassified class of the phylum Nucleocytoviricota (Supplementary Data 4 and 6). About 28% (n = 257) of the 911 viruses were successfully linked to their microbial hosts (by iPHoP[26]) in four archaeal (Halobacteriota, Methanobacteriota, Thermoplasmatota, and Thermoproteota) and 10 bacterial (Actinobacteriota, Bacteroidota, Bdellovibrionota, Campylobacterota, Chloroflexota, Cyanobacteria, Firmicutes, Marinisomatota, Patescibacteria, and Proteobacteria) phyla (Supplementary Data 4). All their hosts contained genes (2 to 89 distinct genes) involved in MMP (Supplementary Data 7). About two-thirds of the host-linked viral contigs (163 of 257) encoded MM AMGs that were also detected in their hosts (Supplementary Data 4).

For each of the 24 MM AMGs, we selected one protein sequence from a highly confident viral contig (Supplementary Fig. S2) as an example to investigate the conserved domain and putative protein structure (see Methods). These in silico analyses revealed that all 24 AMGs exhibited the conserved functional domains and structural configurations (100% confidence for all the tested AMGs, except the *fwdF* gene with 99%; the confidence represents the probability that the match between the studied sequence and the template in the database is a true homology[27]) of their corresponding enzymes (Supplementary Data 5), suggesting that they likely encode functional AMGs. These results indicate that viruses could be largely underexplored players in ecosystem MM.

Investigating the metabolic roles of the 24 MM AMGs, we found that 17 of them could also participate in metabolic pathways other than MM, while the remaining seven AMGs (i.e., *mtrA*, *pmoC*, *fwdF*, *fae*, *cofE*, *cofF*, and *frhB*) exclusively participate in the MMP and thus had a high confidence in supporting direct viral modulations of microbial MM (Fig. 1; Supplementary Data 5, Figs. S1, S3, and S4). Among these seven AMGs, *fwdF* and *fae* were each detected on only one viral contig, while the others were identified on 3 to 25 viral contigs (see Supplementary Information for additional descriptions about viral contigs containing these seven AMGs). Functionally, the *pmoC* gene participates in the aerobic methane oxidation pathway of bacterial methanotrophs[28], while the other six genes *mtrA*, *fwdF*, *fae*, *cofE*, *cofF*, and *frhB* are involved in the pathways of methanogenesis and/or anaerobic oxidation of methane (AOM)[29] (Fig. 1A & Supplementary Fig. S1). The *pmoC* (methane monooxygenase subunit C) gene encodes a subunit of the particulate methane monooxygenase (pMMO) that catalyzes the aerobic oxidization of methane to methanol in bacteria (Fig. 1A & Supplementary Fig. S1)[28]. In the methanogenetic pathway, the *mtrA* (tetrahydromethanopterin S-methyltransferase subunit A) gene encodes a subunit of the membrane-associated multienzyme complex Mtr that transfers the methyl group of $N^5$-methyltetrahydromethanopterin to coenzyme M (CoM) and produces Methyl-CoM[30], which is an exergonic ($\Delta G^{\circ\prime}$ = −29 kJ/mole), sodium-ion-translocating step contributing to ion motive force in the methanogens' energy metabolism. This energy conservation mechanism happens in all methanogens being able to produce methane from $CO_2$ or acetate[31]. The gene product, Methyl-CoM, is essential for the final step of methanogenesis by methanogens[32]. The *fwdF* gene encodes an iron-sulfur protein as the subunit F of formylmethanofuran dehydrogenase, which can catalyze the reduction of methanofuran and $CO_2$ to formylmethanofuran (Fig. 1A and Supplementary Fig. S1), in the first step of methanogenesis from $CO_2$[33,34]. The *fae* gene encodes the formaldehyde activating enzyme (Fae) catalyzing the condensation of formaldehyde with tetrahydromethanopterin (THMPT) to methylene-THMPT[35], an intermediate in methanogenesis from $CO_2$ (Fig. 1A & Supplementary Fig. S1). The remaining three genes *cofE*, *cofF*, and *frhB* are relevant to the synthesis of coenzyme $F_{420}$[36–38], which impacts the production of methylene-THMPT and 5-Methyl-THMPT, also intermediates for methanogenesis from $CO_2$ (Fig. 1A and Supplementary Fig. S1).

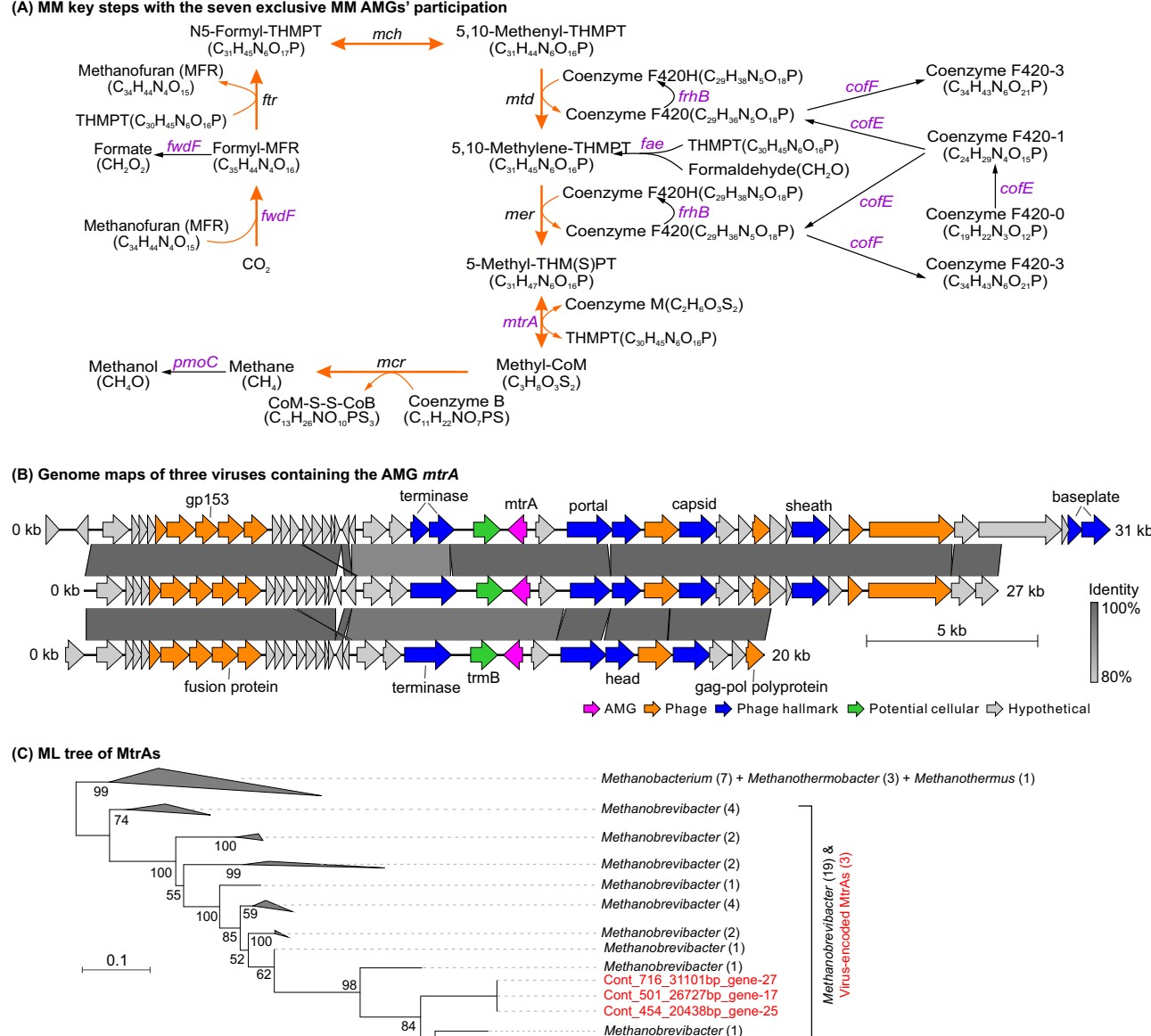

**Fig. 1 | Characterization of exclusive MM AMGs. A** Schematic for viral participations in key MMP steps via encoding seven AMGs that exclusively participate in MMP. Viruses encoded seven AMGs (*fwdF, fae, frhB, cofE, cofF, mtrA,* and *pmoC*; as colored in purple text) to impact the key steps in both methane production and oxidation. The methanogenesis pathway from $CO_2$ to methane is indicated by orange arrows. More information for 17 additional AMGs that could potentially participate in both MMP and other types of metabolism pathways is provided in Supplementary Figs. S1, S2, and Data 5. **B** Genome maps of three viral contigs carrying the AMG *mtrA* gene. The three viral contigs belonged to the same viral population (with 97.4–97.8% genomic identities among each other) and carried an identical *mtrA* gene. CheckV was used to assess host-virus boundaries and remove potential host fractions on the viral contig. Genes were marked by five colors to illustrate AMGs (purple), phage genes (orange), phage hallmark genes (blue),

potential cellular genes (green), and hypothetical protein genes (grey). **C** Phylogenetic tree of the viral and microbial *mtrA* genes. The tree was inferred using the maximum likelihood method with protein sequences. Parametric bootstrap values (expressed as percentages of 1,000 replications) are shown at branching points. The scale bar indicates a distance of 0.1 substitutions per position in the alignment. The viral and microbial MtrA sequences are indicated in red and black, respectively. The numbers in parentheses indicate the number of protein sequences assigned to each group. The full phylogenetic tree (without collapsed groups) is provided in Supplementary Fig. S5A. The genomic maps and phylogenetic trees for the other six exclusive MM AMGs (*pmoC, fwdF, fae, cofE, cofF,* and *frhB*) are provided in Supplementary Figs. S3 and S5B–G. MM, methane metabolism; MMP, methane metabolism pathway.

Phylogenetic analyses of the above seven exclusive MM AMGs suggested that viruses have potentially acquired the *mtrA* genes (n = 3) from methanogens of the genus *Methanobrevibacter* (Euryarchaeota) (Fig. 1C & Supplementary Fig. S5A); the virus-encoded *pmoC* genes (n = 25) have potentially been transferred from methylotrophs belonging to several different genera within the phylum Proteobacteria, including *Methylobacter, Methylomagnum,* and *Methylocystis* (Supplementary Fig. S5B); and the *fae* gene (n = 1) might have been transferred from *Methylophaga* or *Pseudomethylobacillus*

(Supplementary Fig. S5D). The remaining four genes (i.e., *fwdF, cofE, cofF,* and *frhB*) were more divergent from known and taxonomically characterized microbial genes, and thus could not be confidently linked to the potential gene transfer events from hosts to viruses (Supplementary Fig. S5C, E–G).

We assessed the habitats associated with each of the 24 MM AMGs, finding that host-associated samples (i.e., rumen) contained 16, whereas environmental habitats contained between one to seven MM AMGs, including marine water (7 AMGs), marine sediment (5), lake

water (3), lake sediment (1), and hot spring sediment (2) (Fig. 2 & Supplementary Data 5). All 24 MM AMGs were also found on microbial contigs from the same environment where the AMGs were identified (Supplementary Data 2). Surprisingly, we did not find MM AMGs in some of the environmental habitats where we found 138–180 microbial MM genes, such as river estuary sediment, permafrost active layers, and wetland sediment (Supplementary Data 2 and 3). Focusing only on AMGs involved in methane production, we identified 10 genes that can directly participate in or synthesize an intermediate for the pathway of methanogenesis from $CO_2$ or acetate, including six that exclusively participate in MMP (i.e., *mtrA*, *fwdF*, *fae*, *cofE*, *cofF*, and *frhB*; Fig. 1A) and four that could also be involved in other metabolic pathways (*ackA*, *pta*, *cooS*, and *glyA*; Supplementary Fig. S1 & Data 5; see Supplementary Information for their potential roles in methane production pathways); nine of them came from host-associated rumen samples and only one to three were found in the environmental habitats including marine water, marine sediment, lake water, and lake

sediment. Thus, despite representing less than 30% (286/982; Supplementary Data 1 and 3) of the metagenomes analyzed, host-associated samples (i.e., rumen) contained most of the identifiable MM AMGs (including those potentially participating in methane production), which were less common in environmental habitats (e.g., lake sediment, lake water, marine water, and marine sediment) where microbial MM genes (from 138 to 183 distinct genes) were also present. These results suggest that the extent to which viruses use AMGs to modulate host MM processes, including methane production, may vary depending on the habitats in which they dwell.

**Vrana Lake sediment comprises mostly novel viral genera**
Given that MM AMGs were apparently less common in publicly available metagenome datasets from environmental habitats, we adopted a targeted approach to look for additional MM AMGs from methane-rich environmental samples and further explore the impact of viruses on MM via host infection, by generating metagenomes from the methane-

**(A) Predicted hosts of viuses containing MM AMGs**

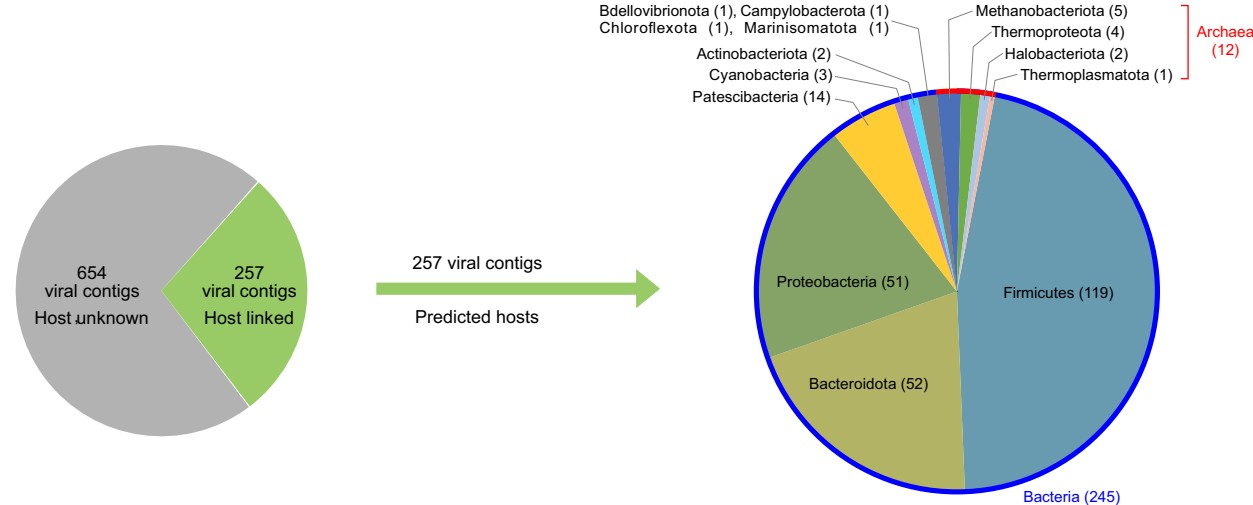

**(B) Habitat association of all 24 MM AMGs**

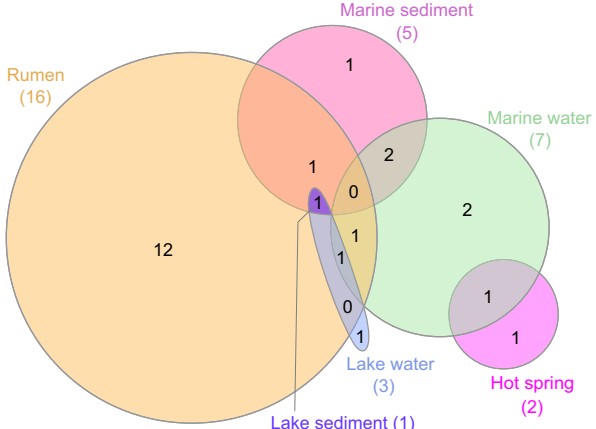

**(C) Habitat association of the 10 MM AMGs involved in the pathway of methanogenesis from $CO_2$ or acetate**

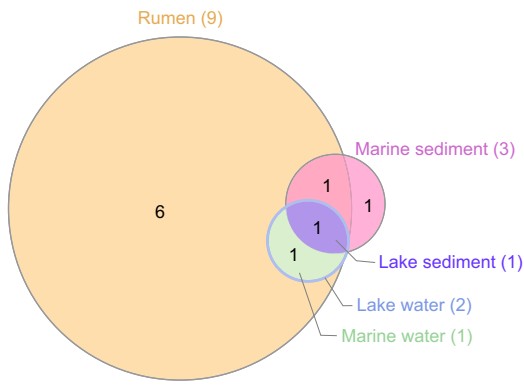

**Fig. 2 | Predicted hosts of viruses encoding MM AMGs and habitat association of MM AMGs. A** Phylum-level host predictions of 257 viruses that encoded MM AMGs. Of the 911 viral contigs encoding MM AMGs, 257 were successfully linked to hosts that belonged to four archaeal and 10 bacterial phyla. Additional information about the predicted hosts is provided in Supplementary Data 4. **B, C** Habitat association of all the 24 MM AMGs (**B**) and the 10 MM AMGs involved in methanogenesis pathway (**C**). We identified 24 distinct MM AMGs from six habitats: rumen (16 AMGs), marine water (7), marine sediment (5), lake water (3), lake sediment (1), and hot spring sediment (2). Seven of these genes were identified in

2–4 habitats, and the remaining 17 were found exclusively in one of these habitats. Of the 24 MM AMGs, 10 genes (i.e., *mtrA*, *fwdF*, *cofE*, *cofF*, *frhB*, *ackA*, *pta*, *cooS*, *glyA*, and *fae*) can directly participate in or synthesize an intermediate for the pathway of methanogenesis from $CO_2$ or acetate (Supplementary Fig. S1). Nine of these 10 AMGs were found in rumen, while only one to three were found from other detectable environmental habitats including marine water, marine sediment, lake water, and lake sediment. Three of these genes were identified in 2–4 habitats, and the remaining six and one were found exclusively in rumen and marine sediment, respectively.

rich sediment of Vrana Lake in Zadar County, Croatia. Six pairs of bulk metagenomes and viromes were constructed for the Vrana Lake sediment (VLS) samples, recovered from two sediment cores at 50, 100, and 225 cm deep below the lake sediment surface. The cores were obtained from two sites within the lake: a *muddy site* consisting of organic-rich sediments within a concave depression (pockmark) of the sediment surface with fluid and gas (e.g., methane) efflux; and a *sandy site* in an area of sandy sediments and no visible pockmark depressions (Supplementary Fig. S6 & Data 8).

We recovered 3,260 viral contigs from the above VLS metagenomes. These contigs were clustered into vOTUs if they shared ≥95% nucleotide identity across 80% of their lengths[39], resulting in 3,146 vOTUs (≥5 kb), including 1,050 "long" (≥10 kb) vOTUs (Supplementary Data 8). Taxonomic analyses, by comparing VLS viruses to viral genomes in both the NCBI RefSeq and IMG/VR databases (see Methods), revealed that most of the VLS long vOTUs (911 of 1,050) could not be taxonomically classified, indicating a high degree of novelty among VLS viruses. The remaining 139 vOTUs were assigned to Caudoviricetes, Faserviricetes, and Megaviricetes (Fig. 3A; Supplementary Fig. S7 & Data 9).

## Viral communities differ between sediment sites and across sediment depths

The cellular microbial communities, investigated based on relative abundances of the 99 bacterial/archaeal metagenome-assembled genomes (MAGs) recovered from VLS metagenomes (Supplementary Data 10; see Methods), were distinct between muddy and sandy sites (Supplementary Fig. S8), which had very different physicochemical conditions (e.g., the total nitrogen and dissolved organic carbon were 8.9 and 2.3 times higher, respectively, in the muddy vs. sandy sediment; Supplementary Data 8). Similarly, the muddy and sandy sampling sites comprised mostly different viruses, with only 4.2% (131 of 3,146) of VLS vOTUs shared between sites (Fig. 3B). Ordination analysis, using the relative abundance data of vOTUs (Supplementary Data 11), confirmed that viral communities were significantly (p = 0.015) different between sites (Fig. 3C). Viral communities also varied with depth (i.e., 50, 100, and 225 cm deep), with only 5.7% (181 of 3,146) of vOTUs detected in samples from all three depths and the majority (75.0%; 2,360 of 3,146) of vOTUs being unique to a single depth (Supplementary Fig. S9A). Additionally, a comparison between bulk metagenomes and viromes found that 97.2% of VLS vOTUs (3,058 of 3,146) were retrieved exclusively from bulk metagenomes, suggesting that most of the recoverable VLS viruses might be within the cellular fraction captured by bulk metagenomes rather than in the viral particle fraction, though our data could not eliminate the possibility that the viromes might have only captured a subset of VLS extracellular viruses (e.g., some extracellular viruses might have been adsorbed to the sediment particles which were removed from viromes via filtering, but captured in bulk metagenomes) (Supplementary Information; Supplementary Fig. S9B & Data 11). For maximizing the virus recovery, we combined viruses identified from both bulk metagenomes and viromes for all further analyses.

## Abundant viruses likely infect dominant microbes of the Thermoproteota and Chloroflexi to impact the sediment ecosystems

To explore the potential viral impacts on VLS ecosystems, we investigated virus-host linkages as reported previously (e.g., in soil and seawater[13,40]), via the iVirus tool VirMatcher[41] that aggregates four different methods for host predictions (see Methods). Using the 99 VLS bacterial/archaeal MAGs as the host database (Supplementary Data 10; See Methods), we could link 2,167 of the 3,146 vOTUs (68.9%) to microbial hosts belonging to 17 different phyla (Fig. 4A; Supplementary Data 12). The VLS microbial communities were dominated by Thermoproteota (relative abundance: average 24.7% and range 12.7–41.4%; archaea) and Chloroflexi (average 23.5% and range

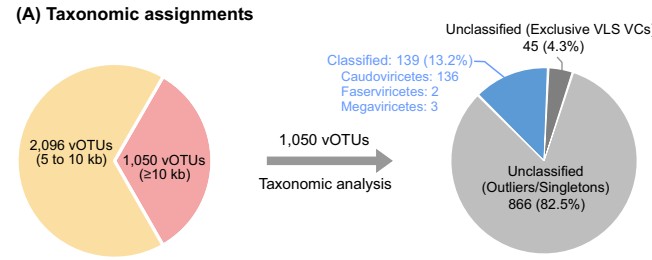

**(A) Taxonomic assignments**

2,096 vOTUs (5 to 10 kb)    1,050 vOTUs (≥10 kb)

1,050 vOTUs → Taxonomic analysis

Classified: 139 (13.2%)
Caudoviricetes: 136
Faserviricetes: 2
Megaviricetes: 3

Unclassified (Exclusive VLS VCs) 45 (4.3%)

Unclassified (Outliers/Singletons) 866 (82.5%)

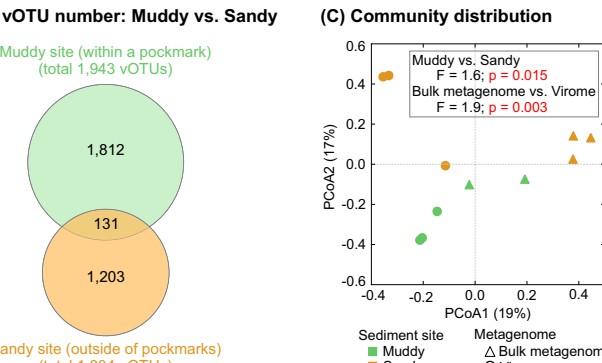

**(B) vOTU number: Muddy vs. Sandy**

Muddy site (within a pockmark) (total 1,943 vOTUs)
1,812
131
1,203
Sandy site (outside of pockmarks) (total 1,334 vOTUs)

**(C) Community distribution**

Muddy vs. Sandy
F = 1.6; p = 0.015
Bulk metagenome vs. Virome
F = 1.9; p = 0.003

PCoA2 (17%) vs PCoA1 (19%)

Sediment site: Muddy (green), Sandy (orange)
Metagenome: Bulk metagenome (triangle), Virome (circle)

**Fig. 3 | Viral communities of Vrana Lake sediments (VLS). A** Taxonomic assignments of VLS vOTUs. The left chart shows the fraction of "long" vOTUs (length ≥10 kb) among all VLS vOTUs (n = 3,146). The right chart shows the taxonomy of VLS "long" vOTUs, when compared to viral genomes in the NCBI RefSeq and IMG/VR databases. Further details of taxonomic results are listed in Supplementary Data 9. **B** Shared and unique vOTUs between the two sediment sites (Muddy vs. Sandy sites) in Vrana Lake as shown in Supplementary Fig. S6. Only 4.2% of the 3,146 vOTUs were presented in both sites, while the remaining 95.8% were unique in either site. **C** PCoA plot of VLS samples based on the relative abundances of vOTUs. The relative abundances of all vOTUs are provided in Supplementary Data 11, and the source data for the Bray Cutis distance matrix are provided as a Source Data file. Samples are marked by the two sediment sites (Muddy and Sandy sites in green and orange, respectively) and the two metagenome types (bulk metagenome and virome as triangles and circles, respectively). The differences of viral communities between both sampling sites and metagenome types were assessed by PERMANOVA (Permutational Multivariate Analysis of Variance; permutations = 999) tests. The p values < 0.05 are indicated in red.

17.9–29.4%; bacteria) (Supplementary Fig. S10). We then calculated lineage-specific virus/host abundance ratios to assess viral infections for specific phyla and found that the most abundant VLS viruses were predicted to infect the above two most dominant microbial phyla, Thermoproteota and Chloroflexi (Fig. 4B; Supplementary Fig. S10). A substantial portion of VLS vOTUs were also linked to Desulfobacterota (Fig. 4A), which, however, was present at low levels or absent across the samples in terms of the identifiable MAG relative abundances (Supplementary Fig. S10). Some MAGs of Thermoproteota contained genes encoding for key steps of MM[42–45], while some members of Chloroflexi are aerobic methanotrophs[46,47] or are able to reduce sulfate to benefit methanogens/methanotrophs via syntrophic interactions[48,49]. In our data, we recovered 23 VLS MAGs belonging to the Thermoproteota, and each of them contained 24–102 (average 65) genes involved in MMP (Supplementary Data 13). Overall, these findings showed that the VLS viruses infected dominant microbial phyla, including ones involved in MM, and thus likely had an important impact on the sediment ecosystems.

## Some Vrana Lake sediment viruses encode genes to modulate host metabolisms

The 99 VLS MAGs contained a total of 5,503 genes (136 distinct genes after dereplication) involved in MMP, including genes that can impact the key steps of MM such as the genes encoding

**(A) Predicted hosts**

2,167 vOTUs (Host linked)

Total 3,146 vOTUs

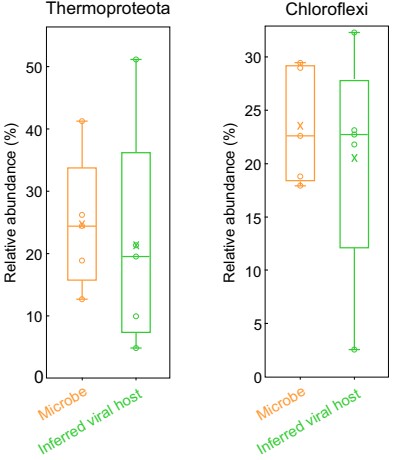

**(B) Host and virus relative abundances for the two most abundant phyla**

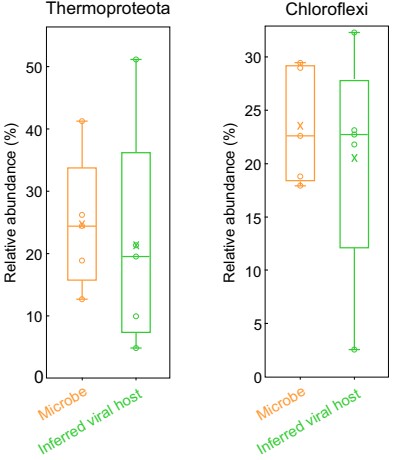

**(C) Genome map of the vOTU "S225_M_175_17980bp" encoding the AMG *bfr***

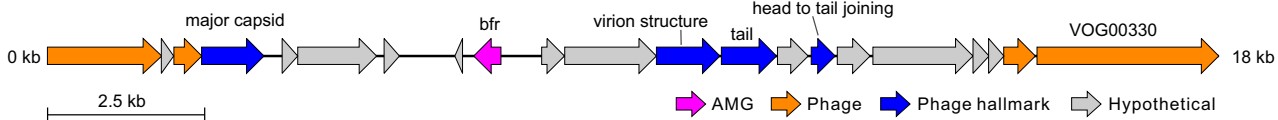

**(D) ML tree of Bfrs**

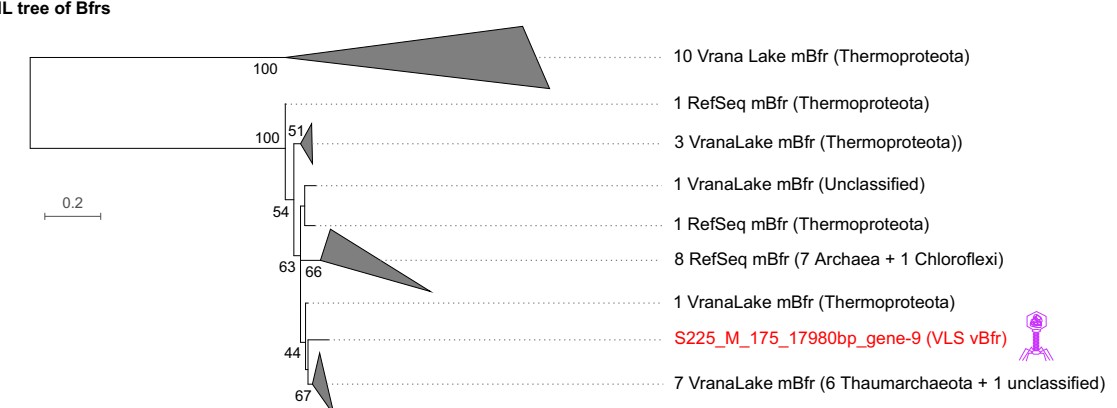

- 10 Vrana Lake mBfr (Thermoproteota)
- 1 RefSeq mBfr (Thermoproteota)
- 3 VranaLake mBfr (Thermoproteota))
- 1 VranaLake mBfr (Unclassified)
- 1 RefSeq mBfr (Thermoproteota)
- 8 RefSeq mBfr (7 Archaea + 1 Chloroflexi)
- 1 VranaLake mBfr (Thermoproteota)
- S225_M_175_17980bp_gene-9 (VLS vBfr)
- 7 VranaLake mBfr (6 Thaumarchaeota + 1 unclassified)

**Fig. 4 | VLS virus-host interactions. A** Predicted phylum-level hosts of VLS viruses. Of the 3,146 vOTUs, 2,167 were linked to putative hosts, with 1,110 and 1,057 vOTUs infecting bacteria and archaea, respectively. The number of vOTUs putatively infecting each phylum is indicated in parentheses after the phylum name. The six bacterial phyla include of Patescibacteria (5), Zixibacteria (3), Armatimonadota (1), Aureabacteria (1), Bacterioidota (1), and OLB16 (1). **B** Relative abundances of the two most abundant VLS microbial phyla and their predicted viruses. The two phyla are Thermoproteota (left panel of box plots) and Chloroflexi (right panel of box plots). Relative abundances of microbes and vOTUs were obtained based on their coverages generated by read mapping to MAG populations and vOTUs. Box-plot elements: center line, median; x symbol, mean; circle, individual data point (n = 5

for each of the box plots); box range, upper and lower quartiles; whiskers, data range. Source data are provided as a Source Data file. **C** Genome map of the vOTU S225_M_175_17980bp encoding the AMG *bfr*. The genome content was characterized by the same methods described in Fig. 1B. **D** Phylogenetic tree of the viral and microbial *bfr* genes. The tree was inferred using the maximum likelihood method with protein sequences (see Methods). Parametric bootstrap values (expressed as percentages of 1,000 replications) ≥40 are shown at branching points. The scale bar indicates a distance of 0.2 substitutions per position in the alignment. The viral Bfr sequence is indicated in red and other sequences are indicated in black. The full phylogenetic tree (without collapsed groups) is provided in Supplementary Fig. S11. VLS, Vrana Lake sediment.

methylenetetrahydromethanopterin dehydrogenase (Mtd), 5,10-methylenetetrahydromethanopterin reductase (Mer), tetrahydromethanopterin S-methyltransferase (Mtr), and methyl-coenzyme M reductase (Mcr) (Supplementary Data 13). To assess if VLS viruses also encode MM AMGs and thus could be modulating the hosts' MM, we annotated genes for all the VLS vOTUs (Supplementary Data 14) and screened them for putative virus-encoded AMGs, including MM AMGs. After rigorous inspection (see Methods),

we identified 13 putative AMGs from 11 VLS vOTUs (Supplementary Data 15). Interestingly, none of these AMGs were predicted to be directly involved in the MMP, which would agree with the inference of our analysis of publicly available metagenomes that the extent to which viruses modulate hosts' MM may vary by habitats, and that MM AMGs seem to be less common in environmental habitats including lake sediments (<2% of the publicly available lake-sediment metagenomes had ≥1 MM AMG).

However, the fact that we identified 13 AMGs suggests that VLS viruses do still have the potential to modulate host metabolism in energy, carbohydrates, amino acids, nucleotides, cofactors, and vitamins (Supplementary Data 15). Particularly, we identified a vOTU that was predicted to infect a putative methanogenic Bathyarchaeia (a class of the phylum Thermoproteota) and which encoded a Thermoproteota-derived AMG for bacterioferritin (*bfr*) that oxidizes $Fe^{2+}$ to $Fe^{3+}$[50] (Fig. 4C, D; Supplementary Fig. S11; Supplementary Information). Evolutionary pressure assessments within species and across lineages found that this virus-encoded Bfr was likely functional and under purification selection ($pN/pS = 0$; average $dN/dS = 0.114$; Supplementary Data 15 and 16). Iron is essential for numerous metabolic processes[51], including microbial MM[52–54]. These results suggest that this virus might have the potential to modulate iron metabolism of a methanogenic host and thus indirectly impact MM in VLS (see Supplementary Information for additional discussion).

In silico analyzes suggested that the 13 AMGs detected are likely functional. All of them had conserved functional domains (Supplementary Data 15), and when their protein sequences were structurally modeled using Phyre2[27], they had 100% confidence scores to their closest template proteins (Supplementary Data 15 and S10). Furthermore, microdiversity analyzes found that the $pN/pS$ values, a proxy for gene selection pressure[55,56], were <1 for all the testable VLS AMGs, suggesting that they were under purifying selection (Supplementary Data 15). While no AMGs that can directly participate in MMP were detected from these methane-rich lake sediments, these results indicate that VLS viruses encode functional AMGs that likely alter microbial metabolisms in the Varana Lake sediments, including an AMG that have an indirect influence on MM through manipulating a putative methanogen's iron metabolism.

## Discussion

After carbon dioxide, methane is the second largest contributor to warming, accounting for approximately 20% of greenhouse gas-driven warming[3–5]. While it is widely accepted that bacteria and archaea are major players in the global methane cycling, little was known about how viruses might impact MM. This study identified 24 virus-encoded MM AMGs in 911 viral contigs by analyzing 982 published metagenomes from environments where microbial MM is known to occur, and where microbial genes involved in MMP were detected. We found that the extent to which viruses use MM AMGs to modulate host MMP may vary depending on the ecological properties of the habitat in which they dwell. Specifically in lake sediments, less than 2% of the publicly available metagenomes contained ≥1 MM AMG and no MM AMG was identified from the 11 metagenomes of Vrana Lake sediments, in which methane emission has been detected. This finding is consistent with previous reports of the habitat-specific association of AMGs in the environments[15,57].

Other than the seven exclusive MM AMGs, among the 24 MM AMGs, the remaining 17 could also be involved in other metabolic pathways, and thus might not be directly related to MM (Supplementary Data 5). For example, the carbon monoxide (CO) dehydrogenase gene (*cooS*; identified from a rumen metagenome; Supplementary Data 5) catalyzes the oxidation of CO to $CO_2$[58], which is the substrate of a methanogenesis pathway from $CO_2$ in ruminants[59]. In addition, the oxidation of CO to $CO_2$ in itself could be a step of methanogenesis using CO as the substrate[60] and an energy generating metabolic reaction[61]. However, the $CO_2$ produced may not be exclusively used for methane metabolism. Thus, while many of the identified AMGs have the potential to participate in MM, without further verification, their actual functions remain hypothetical. Notably, we did not identify virus-encoded *mcr* genes in this meta-analysis. The *mcr* genes encode for the methyl-coenzyme M reductase (MCR), a key enzyme of MM, catalyzing the final step of methanogenesis and the first step of anaerobic oxidation of methane to achieve methane production and oxidation, respectively[62]. Interestingly, while not yet found in viral genomes, the *mcr* genes were recently discovered in a novel group of extrachromosomal elements called "Borgs", that are associated with ANME in the genus *Methanoperedens*[16]. While biologically, viruses could possibly acquire *mcr* genes from microbes, like they acquired other AMGs, the *mcr* genes may not have been detected in our analyses because: (i) they might belong to rare viral species not captured by our sequencing; (ii) viruses might carry *mcr* genes that were highly similar to those in hosts' genomes, precluding the accurate assemblies and identification of viral-encoded *mcr* genes; or (iii) *mcr* genes might not be beneficial for viral survival and have therefore not been maintained, in accord with the fact that so far no virus-encoded *pmoAB* or *amoAB* genes were found, despite the existence of virus-encoded *pmoC* and *amoC* genes[9,12,13]. A more definitive answer on the presence or absence of virus-encoded *mcr* genes, among other genes encoding for key steps of MM that have not been discovered on viral contigs thus far, might be possible with deeper sequencing effort, as sequencing costs decline, and improved assemblies via long-read viromics[63] and/or viral binning[64]. In future studies, these developments will enable us to further expand our view of the viral impacts on methane cycling.

Overall, these findings consolidate our understanding on how viruses might modulate methane production and oxidation via predating host populations and modulating hosts metabolism. They also suggest that the extent to which viruses use AMGs to modulate host MM processes may vary by the habitats in which they dwell, a pattern that may be replicated for viral modulation of other metabolic processes. Future studies are necessary to experimentally validate the proposed host modulation by examining the activity and functionality of virus-encoded proteins of some key AMGs and to further test the presence pattern of MM AMGs and its mechanism as more host-associated and environmental metagenomes become available. Since microbes are key players of methane production and oxidation, the insights gained here reinforce the so far limited knowledge of viral contributions to MM and perhaps climate warming and raise the necessity for including viruses in future ecosystem and geochemical models of MM.

## Methods
### Published metagenome analyses
To investigate how viruses might modulate hosts' metabolic processing to participate in methane cycling, we analyzed 982 publicly metagenomes from both host-associated and environmental habitats that contained 138–183 genes involved in MMP (Supplementary Data 1, 2, and 3; including rumen, marine water, marine sediment, lake water, lake sediment, river estuary sediment, hot spring, and permafrost active layers, among the 15 habitats). Depending on data availability (indicated in Supplementary Data 1), these metagenomes were analyzed by assembling contigs, identifying viral genomes, annotating viral genes, and/or rigorously screening them for putative virus-encoded MM AMGs, as described in below method sections.

### VLS site characterization and field sampling
Two deep sediment cores were collected in 2015 from two sites of the Vrana Lake in Zadar, Croatia: One core was obtained within a pockmark depression in muddy sediment area (muddy site), and a second core was sampled from a sandy sediment area with no visible pockmarks (sandy site; Supplementary Fig. S6 & Data 8). Three sediment samples were collected from each core, at 50, 100, and 225 cm, respectively, below the lake sediment surface of each site. These six sediment samples were frozen at −20 °C once sampled in the field, and then were transported to the laboratory, where they were stored at −20 °C for further analyses, including filtration and DNA extraction.

## Sample processing and genomic DNA isolation

Each sample (0.5 g sediment) was used for bulk DNA extraction with a DNeasy PowerSoil Isolation Kit (Cat No. 12888-100, QIAGEN) according to the manufacturer's instructions. In addition, the extracellular viruses were extracted from each sample (0.9 g sediment) by suspending the sediment using AKC buffer (1% potassium citrate, 1% PBS, and 150 mM $MgSO_4$) by horizontally shaking at 400 rpm for 15 min at 4 °C, according to a previously established protocol[65]. The liquid suspension (about 12 mL) was then passed through a polycarbonate 0.22-μm-pore-size filter (Cat No. GTTP02500, Isopore) to remove cells and particles >0.22 μm. Samples were incubated with 100 U DNase I per 1 mL of sample (Roche) with DNase I reaction buffer (final 10 mM Tric-HCl, 2.5 mM $MgCl_2$, 0.5 mM $CaCl_2$, pH 7.6) at 4 °C for 48 hr. DNase was inactivated by addition of EDTA and EGTA to a final concentration of 100 mM. The virus-like particles in the filtrate were concentrated to 0.5 mL using 100 kDa Amicon Ultra Concentrators (EMD Millipore, Darmstadt, Germany) and preserved at 4 °C until DNA extraction (within 2 hours). Genomic DNA from viral concentrates was isolated using the same protocol as isolating the bulk DNA above. Both bulk and viral DNA were preserved at −20 °C until further processing.

## Metagenomic sequencing

Theoretically, bulk DNA was able to capture all viruses (both intracellular and extra-cellular viruses) from the sediments, while the viral DNA extracted from the filtrates specifically captured the extra-cellular viruses. To maximize viral discovery and gain insight into the proportion of the extra-cellular viruses in VLS, this study analyzed both bulk and viral DNA, which were subjected for bulk and viral metagenome (virome) sequencing, respectively. All metagenomes (i.e., six bulk metagenomes and six viromes) were sequenced at the Joint Genome Institute (JGI), Department of Energy, USA. Briefly, the DNA libraries were prepared using the Nextera® XT Library Prep Kit (Cat No. 15032354, Illumina) and sequenced on the Illumina NovaSeq platform (2 × 150 bp). Sequencing failed for one bulk metagenome sample (i.e., M50_M), which was collected from 50 cm sediment deep of the muddy site core (within a pockmark); thus this sample only had a virome (i.e., M50_V) for further analyzes (Supplementary Data 8).

## Metagenomic read processing and viral identification

Metagenomic data analyses were supported by the Ohio Supercomputer Center, unless stated otherwise. Sequencing reads were filtered for quality by JGI using their previously established standard pipeline[66], generating a total of $9.5 \times 10^{10}$ bases of sequencing data (range $0.3–1.5 \times 10^{10}$ bases, average $8.6 \times 10^{9}$ bases per library; Supplementary Data 8). Then the metagenomic sequence data was assembled to contigs by metaSPAdes[67], using a previously established pipeline for assembling pre-amplified metagenomes (parameters: read deduplication + read error correction + --sc + -k 21,33,55,77,99,127)[68]. The assembled contigs (length ≥5 kb or circular contigs with length 1.5–5.0 kb) from all metagenomes were used for identifying viruses following previously described methods[69], as also described below. Three tools VirSorter v1.1.0[19], DeepVirFinder v1.0[20], and MARVEL v0.2[21] were used for predicting viruses. Contigs were classified as viruses if they met one of the following four criteria: (i) Categories 1, 2, 4, or 5 of VirSorter v1.1.0; (ii) DeepVirFinder score ≥0.9 and p < 0.05; (iii) MARVEL probability score ≥90%; or (iv) DeepVirFinder score ≥0.7 and p < 0.05 and MARVEL probability score ≥70%. Viral contigs identified by the above methods were combined for further analyses.

Viral contigs were first inspected and filtered for potential contaminants by comparing them to viral genomes considered as putative laboratory contaminants (e.g., phages cultivated in our laboratory: *Synechococcus*, *Cellulophaga*, and *Pseudoalteromonas* phages) using Blastn. The remaining contigs were clustered into vOTUs (-species-level taxonomic unit) if they shared ≥95% nucleotide identity across 80% of their lengths[39]. The longest contig within each vOTU was

selected as the seed sequence to represent that vOTU. These efforts generated a total of 3,260 viral contigs, that were clustered into 3,146 vOTUs, including 1,050 "long" vOTUs with length ≥10 kb. The coverages of vOTUs (≥5 kb) were generated using the iVirus' BowtieBatch and Read2RefMapper tools, by mapping quality-filtered reads to vOTUs, and the resulting coverage depths were normalized by library size to "coverage per gigabase of virome" to assess the viral communities in VLS[41,70].

## Taxonomy and ecology analyses

Because viruses lack any single, universally shared gene, we established taxonomy using gene-sharing network analysis from viral sequences ≥10 kb in length using vConTACT v2[24]. Briefly, this analysis compared the 1,050 "long" VLS vOTUs and the 911 public datasets-originated viruses that contained MM AMGs to viral genomes in the National Center for Biotechnology Information (NCBI) RefSeq database (release v201) and the IMG/VR v4 database and generated viral clusters approximately equivalent to known viral genera[13,24,71]. Principal coordinate analyzes (PCoA) were performed using Bray Curtis distance matrices based on the coverage of each vOTU. PERMANOVA (Permutational Multivariate Analysis of Variance; permutations = 999) tests[72] were used to calculate the statistical differences in communities between both sampling sites and metagenome types.

## Microbial genomic analyses

For microbial genomic analyses, quality-controlled reads of the bulk metagenomes were co-assembled using metaSPAdes v3.11.1[67]. The assembled contigs (≥1.5 kb) were then used to bin microbial metagenome-assembled genomes (MAGs), by MetaBat2 v2.12.1[73] using each present binning strategy with and without contig coverage profiles[74]. A total of 99 MAGs, with medium to high quality (completeness ≥40% and contamination ≤10%, via checkM v1.1.10[75]), were generated and then were assigned to a taxonomy using GTDB-Tk v1.3.0[76,77]. Assembly, binning, and quality estimation of prokaryotic MAGs was performed at the Life Science Compute Cluster (https://lisc.univie.ac.at) at the University of Vienna. These MAGs were dereplicated to 83 MAG populations using dRep v1.0.0 with default parameters (sharing ≥95% nucleotide identity across ≥10% of their length)[78]. Metagenomic reads were mapped to MAG populations to characterize their relative abundances using CoverM v0.3.2 with default parameters (https://github.com/wwood/CoverM).

## Viral host prediction

The putative virus-host linkages were predicted in silico using the iVirus tool VirMatcher[41], which aggregates four different methods to provide a statistical confidence score for each host prediction and these methods are based on: (i) tRAN match, (ii) nucleotide sequence composition, (iii) nucleotide sequence similarity, and (iv) CRISPR spacer match. Since viral host prediction benefits from the database that contains microbial genomes from the same ecosystems as viruses[40], we used the microbial MAGs (n = 99), that were recovered from the VLS bulk metagenomes described above, as the microbial database for linking the VLS viruses to their hosts. A summary of the host predictions is available in Supplementary Data 12. The lineage-specific virus/host abundance ratios at phylum level were assessed by comparing the relative abundances of microbial phylum and viruses infecting each phylum[40].

## Virus-encoded AMG identification

The putative AMGs were identified and evaluated for viruses recovered from both 982 publicly published metagenomes from a range of environments where microbial MM genes were detected (see next paragraph) (Supplementary Data 1, 2, and 3) and the 11 VLS metagenomes originally constructed in this study, according to our previously established methods[79]. Specifically, once viral contigs were recovered

from metagenomes, they were processed with VIBRANT to obtain gene functional annotations against the KEGG and PFAM databases and identify putative AMGs by the default parameters[22]. To obtain high-quality and rule out false-positive AMGs from microbial contamination, CheckV (with default parameters, v0.3.0) and manual inspection were then used to assess host-virus boundaries and remove the potential host fraction of the viral contigs[80]. Only AMGs that were surrounded by phage genes, did not contain transposon regions, and had consistent annotations between the KEGG and PFAM databases were included for further analyzes. Metabolism categories of AMGs, including those participating in MMP, were summarized based on KEGG annotations and the pathway modules[18].

For the 982 published metagenomes, we used their preexisting viral contigs for AMG recovery if the data was publicly available, and otherwise de novo assembled the metagenomes ($n = 265$), recovered viral contigs, and/or identified putative AMGs (the data type used for each of the 982 metagenomes is indicated in Supplementary Data 1), using the methods described in the preceding sections. The KEGG IDs of AMGs were used for extracting the genes that could be involved in the MMP, resulting in a discovery of 24 distinct AMGs (on a total of 911 viral contigs containing ≥1 MM AMG) that could participate in 25 steps of the MMP, including seven genes that exclusively participate in MMP (Supplementary Data 5 & Fig. S1). Hosts of the 911 viral contigs containing ≥1 MM AMG were predicted by iPHoP[26] (confidence score ≥90%), resulting successfully virus-host linkages for 257 viral contigs (Supplementary Data 4). Of the 24 AMGs, 17 were detected from only one environment type, while the remaining seven were found in two to four environment types; similarly, six AMGs were identified on only one viral contig and the other 18 AMGs were found on two to 60 viral contigs, except one AMG *glyA* that presented on 642 viral contigs (Supplementary Data 5). For each of the 24 AMGs, we used one viral contig/genome (a contig with a highest viral quality score was selected as the representative, via CheckV's assessment, if the AMG was presented on more than one contig) to illustrate the viral genomic context and AMG position (Supplementary Data 5 & Fig. S2). In addition, the AMGs of the selected viral contigs were further used as examples for analyzing conserved domains by comparing to the domains in the public Conserved Domain Database (CDD v3.20) via NCBI CD-Search[81] and predicting three-dimensional protein structures by Phyre2[27] (Supplementary Data 5). The assembled contigs, excluding viral contigs, were annotated by DRAM against the KEGG and PFAM databases by the default parameters and further used for recovering microbial MM genes based on their KEGG and PFAM annotations (with consistent annotation in the two databases) and the KEGG MM pathway modules[18].

Visualization of the genome maps for the viruses was performed using Easyfig v2.2.5[82]. Phage genes, hallmark genes, and potential cellular genes were identified by VIBRANT, CheckV, and VirSorter[19,22,80,83]. Protein sequences from the AMGs were structurally modeled using Phyre2[27] in normal modeling mode to confirm and further resolve functional predictions. The seven exclusive MM AMGs (*mtrA*, *pmoC*, *fwdF*, *fae*, *cofE*, *cofF*, and *frhB*) and the VLS AMG *bfr* were subjected to phylogenetic analyses to infer its evolutionary history. DIAMOND (v2.0.15) BLASTP[84] was used to query the gene's amino acid sequence against the NCBI RefSeq database (release v214) in a sensitive mode with default settings, to obtain the top 40 hits (top 20 hits if an AMG was identified on more than two viral contigs) as the reference sequences. In addition, microbe-encoded *bfr* genes were extracted from the VLS microbial metagenomes to study possible gene transfers between viruses and their microbial hosts. Multiple sequence alignment was performed using MAFFT (v.7.017)[85] with the E-INS-I strategy for 1000 iterations. The aligned sequences were then trimmed using TrimAl[86] with the flag gappyout. The substitution model was selected by ModelFinder[87] for accurate phylogenetic analysis. Phylogenies were generated using IQ-TREE[88] with ultrafast 1,000 bootstrap replicates,

and then visualized in iTOL (v5)[89]. Potential recombination among genes was evaluated using nine programs: RDP[90], GENECONV[91], BootScan[92], MaxChi[93], Chimaera[94], SiScan[95], LARD[96], Phylpro[97], and 3Seq[98] within RDP5 (v5.23)[99]. A Bonferroni correction with a $p$ value cut-off of 0.05 was applied in each of the tests. A sequence was considered as a true recombinant if supported by at least four of the nine programs. The selection pressure ($pN/pS$) of VLS AMGs were calculated by recruiting VLS metagenomic reads to the AMG-containing vOTUs and identifying the SNPs on AMGs, using the tool MetaPop v1.0 through default parameters[56]. For the VLS AMG *bfr*, branch and site selection pressure ($dN/dS$) analysis across lineages was carried out using codon models with maximum likelihood estimated with the codeml package in PAML (v4.9)[100] (Supplementary Data 16).

## Reporting summary

Further information on research design is available in the Nature Portfolio Reporting Summary linked to this article.

## Data availability

All metagenomic data of VLS samples are newly generated in this study and are available to public via the NCBI Sequence Read Archive (SRA) database with the BioSample accession codes SAMN12796108, SAMN14514859, SAMN14515366, SAMN14515583, SAMN14515785, SAMN15738573, SAMN18258200, SAMN18258201, SAMN18259037, SAMN18259401, and SAMN18261530. All the above accession codes are also provided in Supplementary Data 8. All the analyzed VLS viral contigs and MAGs, as well as the 911 public data-derived viral contigs containing MM AMGs are available at Figshare: https://doi.org/10.6084/m9.figshare.23614812[101]. The accession information of publicly available metagenomes used in this study are provided in Supplementary Data 1. Source data are provided with this paper.

## Code availability

The custom scripts used for analyzing data are available at GitHub: https://github.com/zhiping393/MM[102].

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

## Acknowledgements

This work was supported by the project STIM–REI (Contract Number: KK.01.1.1.01.0003) funded by the European Union through the European Regional Development Fund — the Operational Programme Competitiveness and Cohesion 2014-2020 (KK.01.1.1.01), by DNKVODA project (Contract Number: KK.01.2.1.02.0335), by the Croatian Science Foundation (HRZZ IP-2020-02-9021) to SO, by the U.S. Department of Energy Joint Genome Institute CSP project #503428 to MBS, and partly supported by the Byrd Polar and Climate Research Center Postdoctoral Fellowship and a Heising-Simons Foundation award (2022-4014) to ZPZ, and a Gordon and Betty Moore Foundation Investigator Award (#3790), an NSF Advances in Biological Infrastructure Award (#1759874), and an NSF Biological Oceanography Award (#1829831) to MBS. A portion of this research was performed under the JGI-EMSL Collaborative Science Initiative and used resources at the DOE Joint Genome Institute and the Environmental Molecular Sciences Laboratory, which are DOE Office of Science User Facilities. Both facilities are sponsored by the Office of Biological and Environmental Research and operated under Contract Nos. DE-AC02-05CH11231 (JGI) and DE-AC05-76RL01830 (EMSL). We want to thank Slobodan Miko and Nikolina Ilijanić from the Croatian Geological Survey for their support to VLS sampling, and Natalie Solonenko for DNA extraction. We also appreciate the help provided by Carlos Iniguez, Mohamed M. Mohamed, and Jiarong Guo with helpful discussion, by Yuan Zhou with figure modification, and by Andrew Jermy with manuscript commenting and revising. The selection pressure (*dN/dS*) analyzes benefitted from ZPZ's attending the National Science Foundation-sponsored Polar Genomics Workshop in 2022 (Grant #1935635 and #1935672).

## Author contributions

Z.P.Z., P.P., S.O., and M.B.S. conceived and designed the research. MBS supervised this work. Z.P.Z. analyzed sequencing data. P.P. and S.O. coordinated sampling efforts. J.D. contributed to collecting the lake-sediment public metagenomes. S.K. assembled and binned the VLS MAGs. Z.P.Z. wrote and P.P., S.O., and M.B.S. critically revised the manuscript. All authors revised and approved the final manuscript to be published.

## Competing interests

The authors declare no competing interests.
