## [Peer Review File · Nature Communications]

REVIEWER COMMENTS

Reviewer #1 (Remarks to the Author):

Zhong et al. describe a comprehensive computational search for viral genes involved in methane metabolism as well as an equivalent computational search in a new metagenomic dataset from methane-rich lake sediments. This is an interesting scientific question. The methods the authors use to assemble and identify viral genomes in the metagenomic assemblies are robust. Fundamentally, their primary conclusion, the report of 24 viral AMGs involved in methane metabolism, is most likely accurate. However, crucial analyses of these viral genomes are missing, and there are some organizational issues in the manuscript. Specifically:

1. The authors need to provide an analysis of what the likely hosts are of the viruses encoding viral AMGs. What general classes of phage are these viruses? What are their likely or possible hosts? Do their hosts have genes involved in methane metabolism? Can the authors use a broad host-prediction tool like iPhoP to aid in these predictions?
2. Some conclusions by the authors consist of comparing the distribution of viral AMGs across environments expected to have microbial methane metabolism. But can they quantify the presence of prokaryotic methane pathways in these environments to shed light on this? How do viruses encoding methane metabolism genes correlate with prokaryotes that do so? Can a virus potentially provide a gene otherwise entirely missing from a host, or are they augmenting existing host metabolism? These are the pressing questions that are left unanswered by the manuscript.

Some specific comments:

Abstract: the abstract should focus a bit more on what the viruses encoding the MM AMGs are: what types of viruses encode these genes? What organisms might they infect? This information is missing from the abstract.

Line 84: publicly available metagenomes from environments highly likely to have active biological MM can the authors explain more about how they evaluated whether a metagenome fit this criteria?

Line 87: "isolating viral genomes" can the authors adjust terminology (e.g. "identifying" viral genomes) and briefly mention the methodology used here?

Line 88-90: More details should be in the results describing how MM genes were identified. What qualifies as an MM gene? How were they identified? Just the most basic description of the methods should go here so that a reader does not have to skip to the methods to understand what is actually being reported.

Line 95: Please specify what the "confidence" reported here actually is and what it implies.

Line 138-140: "Where we expect active biological MM..." So, were prokaryotic MM genes present in these environments? How do viral MM genes correlate with the presence of prokaryotic MM genes, and/or prokaryotes involved in methane metabolism? This crucial comparison is missing from the work.

For example, in a later section, it is noted that "The VLS microbial communities were dominated by Thermoproteota" and "Many Thermoproteota members are archaeal methanogens" ... but from the MAGs can the authors identify genes for methanogenesis? What about the methanotrophs?

Reviewer #2 (Remarks to the Author):

Microorganisms are main players in the cycling of the greenhouse gas methane, but recent discoveries of extrachromosomal elements associated with these microbes suggest that they influence the cycling of CH₄, too. The authors Zhong et al. now analyze 982 publicly available metagenomes and generate 11 new metagenomes from a freshwater lake in Croatia, to identify viruses associated with methane metabolizing microorganisms. They find viruses that seem to infect methanogens and aerobic methanotrophs in the existing and their own datasets and then analyze the AMGs encoded on these viruses/phages. This is a generally very interesting topic and the workflows described seem well-chosen. Yet some of the data presented seems a bit disconnected and preliminary for publication. For example, it is not surprising that viral communities differ between sediment sites because the microorganisms differ between those sites. Thus it would be important to resolve the microbial community of the sites or at least present those microorganisms that are involved in methane cycling in these sites (and present this data), and then present the viral MAGs found in those sites. It is also not clear how novel the viral contigs are since they were only compared to cultivable viruses and a selection of metagenomic datasets. There is also no clear boundary between methanogenic archaea and methanotrophic bacteria. A reader that is no expert in this would likely be very confused as everything is presented together. This manuscript unfortunately requires a major revision before it can be published.

Major comments:

- Headers of sections are imprecise. E.g. "Viruses encode 24 AMGs modulating..." suggests this is a general finding. It would be better to rephrase to "Some viruses encode genes that could augment/modulate microbial methane metabolism". Generally, abbreviations should be avoided in results sections because the reader will be lost..
- L: 100-110: The aerobic methane oxidation should be more clearly separated from (reverse) methanogenesis, since these are very distinct processes mediated by completely different microbes
- L111f: what is cool about the finding that mtrA is encoded on a viral contig is that the Mtr complex is directly involved in energy conservation. This part could be really improved by diving in a bit deeper into the function and stating this!
- L169: why were the vOTUs only compared to cultivable viruses? This of course will lead to the (likely false) inference that they are all novel... There is a huge database of viruses/phages derived from metagenomics (and were thus not cultivated) and the viral contigs should be compared to those? The authors then do compare them to "selected published environmental metagenomes"(384) but how and why were these selected?
- L:254: how do the viruses alter the ecophysiology of the sediments? That seems like a really huge claim with no evidence to support this
- L259-275: This reads like a summary or even abstract and not like a discussion.

- L284-287: where does this hypothesis come from and how does the data presented in this paper connect to it?
- Figure 1A: The representation is extremely hard to follow. It also suggests that pmoC is in one cell (virus ?) with the other enzymes? Gene names are to be written in italics. This Figure should be revised.
- Figure 1C: Is it necessary to show this tree in such a collapsed way? It seems to provide little information. As a sidenote: protein names should be capitalized.
- Figure 2: Is it necessary to show this as a main figure? I am more confused by looking at it than having read the section in the text, unfortunately.
- The Figures generally look a bit unpolished... e.g., Figure 3 (A) and (B) are outside the boxes, where (C) and (D) are inside. There are lots of pie charts and circles in several figures.. Figure 4A why does the green arrow restate in green font what is written in the left pie-chart? Were all 2,167 vOTUs assigned to either bacteria or archaea? This seems surprising that taxonomic classification/linkage is possible for every vOTU found..

Minor comments:

- L28: “methanogenic sediments” seems very colloquial, maybe rephrase to “sediments where methane emissions/consumption has been detected”(or whatever is applicable)
- L90: resulting in the discovery..
- L93L if only one protein sequence was investigated, how can the “conserved domains” be investigated ?
- L104: on only one viral contig (not “from”)
- L106: it should be stated that pmoC is part of the aerobic methane oxidation pw of bacteria
- L107: how do the authors know that these genes are not involved in AOM ? Unless there is evidence for this (e.g., the metagenomic samples these viral contigs originate from, only comprise methanogens, and not ANME). If this was not investigated, it should be added “pathway of methanogenesis or reverse methanogenesis”
- L168: how were the vOTUs established?
- L 211: how were vOTUs linked to microbial hosts? This should be stated in brief in the results.
- L221: I don’t think it is accurate to write “many” Thermoproteota members are methanogens. Also not sure why “archaeal” is required at this point, because all methanogens (we know so far!) are archaeal (this would fit in the beginning, but not here)
- L224: had an important
- L223-5: This seems pretty disconnected. What is the evidence for this or why bring this up here?
- L242: “conserved functional domains of their corresponding enzymes” What does this mean?
- L:251: Why was this pN/pS analysis done?
- L282: “significant” may not be the best term here

- L293: CO is also a substrate for methanogens
- L:294: It is a bit odd to have (n=17) in the discussion section as such specific values usually belong to the results
- L297ff: it should be stated that mcr genes were recently found in “Borgs” – archaeal ECEs that are neither typical viruses nor plasmids

REVIEWER COMMENTS

Reviewer #1 (Remarks to the Author):

Zhong et al. describe a comprehensive computational search for viral genes involved in methane metabolism as well as an equivalent computational search in a new metagenomic dataset from methane-rich lake sediments. This is an interesting scientific question. The methods the authors use to assemble and identify viral genomes in the metagenomic assemblies are robust. Fundamentally, their primary conclusion, the report of 24 viral AMGs involved in methane metabolism, is most likely accurate. However, crucial analyses of these viral genomes are missing, and there are some organizational issues in the manuscript. Specifically:

Response #1: We appreciate your kind words in assessing our methods and primary conclusion and thank you for the constructive criticisms on the original manuscript's shortcomings, which greatly helped us improve our work during revision. In the revised version, we performed additional analyses as requested:

- (1) We identified and added information on the putative hosts for these viruses. Please see Response #2 for further details.
- (2) We assigned the taxonomy of these viral genomes by comparing them to viruses from the NCBI RefSeq and IMG/VR databases. Please see Response #2 for further details.
- (3) We assessed the microbial genes involved in MMP in the same environments where we searched for virus-encoded MM AMGs. This assessment helps shed light on the later work about identifying virus-encoded MM AMGs in these environments, and thus greatly improves the contextualization of our data. Please see Response #3 for further details.

Please note that all the line numbers cited in our responses are based on the revised manuscript with tracked changes.

1. The authors need to provide an analysis of what the likely hosts are of the viruses encoding viral AMGs. What general classes of phage are these viruses? What are their likely or possible hosts? Do their hosts have genes involved in methane metabolism? Can the authors use a broad host-prediction tool like iPhoP to aid in these predictions?

Response #2: Thank you for these thoughtful suggestions. In the revised manuscript, we used iPhoP to predict hosts for the 911 viral contigs that we detected to encode MM AMGs. The results are provided in Table S4 (Columns Y–AD; a general summary of viral hosts) and Table S7 (full gene annotations of viral hosts). Briefly, 257 of the 911 viruses were successfully linked to their hosts. Notably, all the predicted hosts contained genes (between 2 and 89) involved in MMP, and about two-thirds of the host-linked viral contigs (163 of 257) encoded MM AMGs that were also detected in their hosts (Column AA in Table S4). To contextualize these results, the below sentences were added to the revised main text (Lines 121-128):

“About 28% (n=257) of the 911 viruses were successfully linked to their microbial hosts (by iPhoP²⁵) in four archaeal (Halobacteriota, Methanobacteriota, Thermoplasmata, and Thermoproteota) and 10 bacterial (Actinobacteriota, Bacteroidota, Bdellovibrionota, Campylobacterota, Chloroflexota, Cyanobacteria, Firmicutes, Marinisomatota, Patescibacteria, and Proteobacteria) phyla (Supplementary Table S4). All their hosts contained genes (2 to 89 distinct genes) involved in MMP (Supplementary Table S7). About

two-thirds of the host-linked viral contigs (163 of 257) encoded MM AMGs that were also detected in their hosts (Supplementary Table S4)."

Regarding the comment on “*general classes of phages for these viruses*”, we assigned the taxonomy of viruses that encoded MM AMGs by comparing their genomes to the viral genomes in NCBI RefSeq and IMG/VR databases via the tool vConTACT v2. About 34% of these viruses (308 of 911) could be assigned a taxonomy and all of them belonged to the class Caudoviricetes of the phylum Uroviricota, except one viral contig that belonged to an unclassified class of the phylum Nucleocytoviricota. These results were provided in Table S4 (Column X) and Table S6, as well as described in the main text (Lines 115-121):

“We compared the 911 viral contigs to the viral genomes/contigs from the NCBI RefSeq database (cultivable viral genomes) and IMG/VR database (uncultivated virus genomes from metagenomics)²² using a genome-based network approach (see Methods)^{23,24}. About 34% of these viruses (308 of 911) could be assigned to taxonomy and all belonged to the class Caudoviricetes of the phylum Uroviricota, except one that belonged to an unclassified class of the phylum Nucleocytoviricota (Supplementary Table S4 & Table S6)”

2. Some conclusions by the authors consist of comparing the distribution of viral AMGs across environments expected to have microbial methane metabolism. But can they quantify the presence of prokaryotic methane pathways in these environments to shed light on this? How do viruses encoding methane metabolism genes correlate with prokaryotes that do so? Can a virus potentially provide a gene otherwise entirely missing from a host, or are they augmenting existing host metabolism? These are the pressing questions that are left unanswered by the manuscript.

Response #3: Thanks for your thoughtful comments. We now also annotated the prokaryotic contigs of the publicly available 982 metagenomes, where we searched for viral MM AMGs, and explored the genes involved in MMP. These metagenomes were from 15 environments, in which the prokaryotes contained 138–183 distinct genes (i.e., after dereplication) involved in MMP (see Table S2 & Table S3). All the 24 viral MM AMGs were also detected from prokaryotes, in the same environment where the viral and prokaryotic genes originated. However, the number of viral and prokaryotic MM genes were uncorrelated across these environments (see Table S3). These results were discussed in the revised main text:

Lines 96-104: “To discover new MM AMGs, 982 publicly available metagenomes from 15 environments (Supplementary Table S1; including rumen, marine water, marine sediment, lake water, lake sediment, river estuary sediment, wetland sediment, and permafrost active layers, among others), were analyzed for microbial genes involved in MMP and viral genomes encoding MM AMGs. The assembled contigs excluding viral contigs, were used to identify microbial genes involved in MMP (based on their KEGG and PFAM annotations and the KEGG MM pathway modules¹⁷) and each of the 982 metagenomes contained 138–183 distinct microbial genes involved in MMP (in total 184 distinct genes from all environments after dereplication; Supplementary Table S2 & Table S3).”

Lines 178-186: *“We assessed the habitats associated with each of the 24 MM AMGs, finding that host-associated samples (i.e., rumen) contained 16, whereas environmental habitats contained between one to seven MM AMGs, including marine water (7 AMGs), marine sediment (5), lake water (3), lake sediment (1), and hot spring sediment (2) (Fig. 2 & Supplementary Table S5). All 24 MM AMGs were also found on microbial contigs from the same environment where the AMGs were identified (Supplementary Table S2). Surprisingly, we did not find MM AMGs in some of the environmental habitats where we found 138–180 microbial MM genes, such as river estuary sediment, permafrost active layers, and wetland sediment (Supplementary Table S2 & Table S3).”*

Some specific comments:

Abstract: the abstract should focus a bit more on what the viruses encoding the MM AMGs are: what types of viruses encode these genes? What organisms might they infect? This information is missing from the abstract.

Response #4: We have revised the abstract to include the information on taxonomy and putative hosts of these viruses (Lines 34-40):

“By analyzing viruses ... we expand this to 24 viral MM AMGs identified on 911 viral contigs, including seven genes that are exclusive to MM pathways. About 34% of the 911 viruses could be taxonomically assigned and almost all belonged to the Caudoviricetes; ~28% were successfully linked to their hosts, in four archaeal and 10 bacterial phyla, including Halobacteriota, Methanobacteriota, and Thermoproteota.”

Line 84: publicly available metagenomes from environments highly likely to have active biological MM
can the authors explain more about how they evaluated whether a metagenome fit this criteria?

Response #5: Yes. In the revised version, we added the analyses of microbial genes in these environments (see Response #3). The results suggested that each of these environments contained 138 to 183 distinct microbial genes involved in MMP and they were thus used to search for viral MM AMGs. Overall, we initially selected metagenomes from environments, based on literature knowledge, known to potentially host methane-cycling microbial communities. Further, we annotated their microbial genes and confirmed the presence of microbial MM genes in these environments. We have modified the below text to explain this:

Lines 83-86: “First, we analyzed 982 publicly available metagenomes from a range of environments, which are known from literature to potentially host methane-cycling microbial communities, and in which we were able to confirm the presence of microbial MM genes, to identify virus-encoded AMGs that could be involved in the MM pathway (MMP), including ...”

Lines 96-104: “To discover new MM AMGs, 982 publicly available metagenomes from 15 environments (Supplementary Table S1; including rumen, marine water, marine sediment, lake water, lake sediment, river estuary sediment, wetland sediment, and permafrost active

layers, among others), were analyzed for microbial genes involved in MMP and viral genomes encoding MM AMGs. The assembled contigs excluding viral contigs, were used to identify microbial genes involved in MMP (based on their KEGG and PFAM annotations and the KEGG MM pathway modules¹⁷) and each of the 982 metagenomes contained 138–183 distinct microbial genes involved in MMP (in total 184 distinct genes from all environments after dereplication; Supplementary Table S2 & Table S3).”

Line 87: "isolating viral genomes" can the authors adjust terminology (e.g. "identifying" viral genomes) and briefly mention the methodology used here?

Response #6: We have adjusted the terminology and added a brief description of methods for identifying viral genomes. The modified text reads as (Lines 108-110):

“Viral genomes were also identified from the assembled contigs of these metagenomes, using a combination of three tools: VirSorter¹⁸, DeepVirFinder¹⁹, and MARVEL²⁰ (see Methods).”

Line 88-90: More details should be in the results describing how MM genes were identified. What qualifies as an MM gene? How were they identified? Just the most basic description of the methods should go here so that a reader does not have to skip to the methods to understand what is actually being reported.

Response #7: We have modified these sentences to include a description of the methods for identifying MM genes. They read as (Lines 110-116):

“In the identified viral genomes, we predicted and annotated viral genes and screened for putative virus-encoded MM AMGs using VIBRANT²¹ and manual curation. Particularly, MM AMGs were extracted based on their KEGG annotations and the MM pathway modules¹⁷. After rigorous inspection (see Methods), 911 viral contigs were identified to contain MM AMGs (Supplementary Table S4), resulting in the discovery of 24 distinct AMGs that potentially participate in 25 metabolic reactions in the MMP (Supplementary Table S5, Fig. S1, & Fig. S2).”

Line 95: Please specify what the "confidence" reported here actually is and what it implies.

Response #8: Thank you for pointing out this omission. The confidence represents the probability (from 0% to 100%) that the match between the studied sequence and the template in the database is a true homology. A higher confidence implies a higher accuracy of the protein structure prediction. We have modified the text to include an explanation of the “confidence”, it reads as (Lines 132-136):

“These in silico analyses ... and structural configurations (100% confidence for all the tested AMGs, except the fwdF gene with 99%; the confidence represents the probability that the match between the studied sequence and the template in the database is a true homology²⁶) ...”

Line 138-140: "Where we expect active biological MM..." So, were prokaryotic MM genes present in these environments? How do viral MM genes correlate with the presence of prokaryotic MM genes, and/or prokaryotes involved in methane metabolism? This crucial comparison is missing from the work.

For example, in a later section, it is noted that "The VLS microbial communities were dominated by Thermoproteota" and "Many Thermoproteota members are archaeal methanogens" ... but from the MAGs can the authors identify genes for methanogenesis? What about the methanotrophs?

Response #9: Please refer to "Response #3" for our arguments about the prokaryotic MM genes present in these environments. We have modified this sentence as (Lines 184-187):

"Surprisingly, we did not find MM AMGs in some of the environmental habitats where we found 138–180 microbial MM genes, such as river estuary sediment, permafrost active layers, and wetland sediment (Supplementary Table S2 & Table S3)."

For the comment on "Thermoproteota members": We recovered 23 MAGs belonging to Thermoproteota from VLS, and each of them contained 24–102 (average 65) genes involved in MMP. Although *mcr* genes were previously found from MAGs of the uncultivated Thermoproteota lineages (e.g., Evans et al., 2015. Science; Qu et al., 2022. ISME J), they were not detected on any of the above 23 MAGs. However, these VLS MAGs contained some other MM genes that can impact the key steps of methanogenesis and/or reverse methanogenesis (i.e., anaerobic methane oxidation), such as the genes encoding heterodisulfide reductase (*hdrA1*, *hdrA2*, *hdrB2*, *hdrC2*, and *hdrD*), methenyltetrahydromethanopterin cyclohydrolase (*mch*), methylenetetrahydromethanopterin dehydrogenase (*mtd*, *mtdB*), 5,10-methylenetetrahydromethanopterin reductase (*mer*), tetrahydromethanopterin S-methyltransferase (*mtrA* and *mtrH*), dimethylamine corrinoid protein (*mtbC*), methylamine corrinoid protein Co-methyltransferase (*mtmB*), and trimethylamine corrinoid protein (*mttBC*). These genomic data suggested that at least some of the VLS *Thermoproteota* members were able to participate in MM, though the available data is unable to determine whether they are methanogens or methanotrophs, or both.

In the revised version, we provided the gene annotations of the above 23 MAGs, along with the other 76 VLS MAGs, and highlighted the genes involved in MMP (see Table S13). In addition, we have revised the text as (Lines 281-287):

"Some MAGs of Thermoproteota contained genes encoding for key steps of MM⁴¹⁻⁴⁴, ... In our data, we recovered 23 VLS MAGs belonging to the Thermoproteota, and each of them contained 24–102 (average 65) genes involved in MMP (Supplementary Table S13)."

Evans PN, Parks DH, Chadwick GL, Robbins SJ, Orphan VJ, Golding SD & Tyson GW (2015) Methane metabolism in the archaeal phylum Bathyarchaeota revealed by genome-centric metagenomics. *Science* 350: 434-438. DOI: <https://doi.org/10.1126/science.aac7745>

Ou YF, Dong HP, McIlroy SJ, et al. (2022) Expanding the phylogenetic distribution of cytochrome b-containing methanogenic archaea sheds light on the evolution of methanogenesis. *ISME J* 16: 2373-2387. DOI: <https://doi.org/10.1038/s41396-022-01281-0>

Reviewer #2 (Remarks to the Author):

Microorganisms are main players in the cycling of the greenhouse gas methane, but recent discoveries of extrachromosomal elements associated with these microbes suggest that they influence the cycling of CH₄, too. The authors Zhong et al. now analyze 982 publicly available metagenomes and generate 11 new metagenomes from a freshwater lake in Croatia, to identify viruses associated with methane metabolizing microorganisms. They find viruses that seem to infect methanogens and aerobic methanotrophs in the existing and their own datasets and then analyze the AMGs encoded on these viruses/phages. This is a generally very interesting topic and the workflows described seem well-chosen. Yet some of the data presented seems a bit disconnected and preliminary for publication. For example, it is not surprising that viral communities differ between sediment sites because the microorganisms differ between those sites. Thus it would be important to resolve the microbial community of the sites or at least present those microorganisms that are involved in methane cycling in these sites (and present this data), and then present the viral MAGs found in those sites. It is also not clear how novel the viral contigs are since they were only compared to cultivable viruses and a selection of metagenomic datasets. There is also no clear boundary between methanogenic archaea and methanotrophic bacteria. A reader that is no expert in this would likely be very confused as everything is presented together. This manuscript unfortunately requires a major revision before it can be published.

Response #10: We appreciate your kind evaluation of our research topic and workflows, and thank you for highlighting the weakness about the data presented to help us improve the work. We have substantially revised the manuscript based on the above comments, which should more deeply explain and contextualize the findings in this study. Please note that all the line numbers cited in our responses are based on the revised manuscript with tracked changes. The revisions include:

- (1) We now introduced Vrana Lake sediment (VLS) microbial communities/functions first, before we discussed VLS viral communities/functions. We also added the gene annotations of all the 99 MAGs obtained from VLS and highlighted all genes involved in MMP. Please see Table S13 and the below text:
 - Lines 238-247: *“The cellular microbial communities, investigated based on relative abundances of the 99 bacterial/archaeal metagenome-assembled genomes (MAGs) recovered from VLS metagenomes (Supplementary Table S10; see Methods), were distinct between muddy and sandy sites (Supplementary Fig. S8), which had very different physicochemical conditions ... Similarly, the muddy and sandy sampling sites comprised mostly different viruses, with only 4.2% (131 of 3,146) of VLS vOTUs shared between sites (Fig. 3B). Ordination analysis, using the relative abundance data of vOTUs (Supplementary Table S11), confirmed that viral communities were significantly ($p = 0.015$) different between sites (Fig. 3C).”*
 - Lines 267-275: *“To explore the potential viral impacts on VLS ecosystems, we investigated virus-host linkages as reported previously (e.g., in soil and seawater^{11,39}),*

via the iVirus tool VirMatcher⁴⁰ that aggregates four different methods for host predictions (see Methods). Using the 99 VLS bacterial/archaeal MAGs as the host database (Supplementary Table S10; See Methods), we could link 2,167 of the 3,146 vOTUs (68.9%) to microbial hosts belonging to 17 different phyla (Fig. 4A; Supplementary Table S12). The VLS microbial communities were dominated by Thermoproteota (relative abundance: average 24.7% and range 12.7–41.4%; archaea) and Chloroflexi (average 23.5% and range 17.9–29.4%; bacteria) (Supplementary Fig. S10).”

- Lines 294-302: “The 99 VLS MAGs contained a total of 5,503 genes (136 distinct genes after dereplication) involved in MMP, including genes that can impact the key steps of MM such as the genes encoding methylenetetrahydromethanopterin dehydrogenase (Mtd), 5,10-methylenetetrahydromethanopterin reductase (Mer), tetrahydromethanopterin S-methyltransferase (Mtr), and methyl-coenzyme M reductase (Mcr) (Supplementary Table S13). To assess if VLS viruses also encode MM AMGs and thus could be modulating the hosts’ MM, we annotated genes for all the VLS vOTUs (Supplementary Table S14) and screened them for putative virus-encoded AMGs, including MM AMGs.”

We would like to point out that a more detailed analyses of the microbial data for VLS samples will be used in a different manuscript that is in preparation. In the current paper, we specifically focused on viruses and only presented limited microbial data (e.g., MAG-based community, taxonomy, and genes involved in MMP) that helped reveal virus-microbe interactions in the sediments.

Apart from the revisions for the VLS datasets, we have also conducted additional analyses of microbial MM genes for the public metagenomic datasets where we searched for viral MM AMGs, as per the request of the other reviewer. Then we reorganized the text to introduce the microbial MM genes first, before we documented the viral MM genes in the environments where the public metagenomic datasets originated. Please refer to Response #3 for further details about this revision.

- (2) In the revised manuscript, we included both cultivated (NCBI RefSeq database) and uncultivated viral genomes (IMG/VR v4 database) for assessing viral novelty. Please see Response #14 for further details.
- (3) We have revised the text to clarify whether we refer to methanogenic archaea, methanotrophic bacteria, or anaerobic methanotrophic archaea:
Lines 72-75: “Recently, some viruses have been found to encode *pmoC* and *cofF* as auxiliary metabolic genes (AMGs), with the potential to supplement the aerobic oxidation of methane by their bacterial host in freshwater lakes⁷.”

Lines 86-88: “including those participating in methane production (i.e., methanogenesis by archaea) and oxidation (either by aerobic methanotrophic bacteria or anaerobic methanotrophic archaea).”

Lines 146-152: “Functionally, the *pmoC* gene participates in the aerobic methane oxidation pathway of bacterial methanotrophs²⁷, while the other six genes *mtrA*, *fwfD*,

fae, cofE, cofF, and frhB are involved in the pathways of methanogenesis and/or anaerobic oxidation of methane (AOM)²⁸ (Fig. 1A & Supplementary Fig. S1). The pmoC (methane monooxygenase subunit C) gene encodes a subunit of the particulate methane monooxygenase (pMMO) that catalyzes the aerobic oxidization of methane to methanol in bacteria (Fig. 1A & Supplementary Fig. S1)²⁷.”

(4) We have carefully improved all the main figures. Please see Response #21 for further details.

We hope that these revisions have better contextualized this work for a publication.

Major comments:

- Headers of sections are imprecise. E.g. “Viruses encode 24 AMGs modulating...” suggests this is a general finding. It would be better to rephrase to “Some viruses encode genes that could augment/modulate microbial methane metabolism”. Generally, abbreviations should be avoided in results sections because the reader will be lost..

Response #11: Thank you for these comments. We have changed this section header to “*Some viruses encode genes that could modulate microbial methane metabolism*” based on your suggestion (Lines 94-95). In addition, we modified the below four section headers:

Line 204: Modified from “*Vrana lake sediment comprises mostly novel viral genera and species*” to “*Vrana Lake sediment comprises mostly novel viral genera*”.

Line 237: Modified from “*VLS viral communities differ between sediment sites*” to “*Viral communities differ between sediment sites and across sediment depths*”.

Lines 265-266: Modified from “*VLS viruses impact ecosystem via infecting dominant microbes*” to “*Abundant viruses likely infect dominant microbes of the Thermoproteota and Chloroflexi to impact the sediment ecosystems*”.

Line 292-293: Modified from “*VLS viruses encode AMGs to modulate hosts’ metabolism*” to “*Some Vrana Lake sediment viruses encode genes to modulate host metabolisms*”.

We removed all the abbreviations in the results headers, but kept the abbreviations in the results sections as we respectively disagree with your opinion that “*abbreviations should be avoided in results sections because the reader will be lost*”.

- L: 100-110: The aerobic methane oxidation should be more clearly separated from (reverse) methanogenesis, since these are very distinct processes mediated by completely different microbes

Response #12: Thank you for pointing out this omission. We have modified the text to clearly clarify that the *pmoC* gene is involved in aerobic oxidation of methane in bacterial methanotrophs (Lines 146-152):

“Functionally, the pmoC gene participates in the aerobic methane oxidation pathway of bacterial methanotrophs²⁷, while the other six genes mtrA, fwdF, fae, cofE, cofF, and frhB are involved in the pathways of methanogenesis and/or anaerobic oxidation of methane

(AOM)²⁸ (Fig. 1A & Supplementary Fig. S1). The pmoC (methane monooxygenase subunit C) gene encodes a subunit of the particulate methane monooxygenase (pMMO) that catalyzes the aerobic oxidization of methane to methanol in bacteria (Fig. 1A & Supplementary Fig. S1)²⁷”.

- L111f: what is cool about the finding that mtrA is encoded on a viral contig is that the Mtr complex is directly involved in energy conservation. This part could be really improved by diving in a bit deeper into the function and stating this!

Response #13: Thanks for your suggestion to enhance our work. We agree and have modified the text to document that the enzyme complex Mtr catalyzes an energy conserving step in the MMP (Lines 152-158):

“In the methanogenic pathway, the mtrA (tetrahydromethanopterin S-methyltransferase subunit A) gene encodes a subunit of the membrane-associated multienzyme complex Mtr that transfers the methyl group of N5-methyltetrahydromethanopterin to coenzyme M (CoM) and produces Methyl-CoM²⁹, which is an exergonic ($\Delta G^{\circ} = -29$ kJ/mole), sodium-ion-translocating step contributing to ion motive force in the methanogens’ energy metabolism. This energy conservation mechanism happens in all methanogens being able to produce methane from CO₂ or acetate³⁰.”

- L169: why were the vOTUs only compared to cultivable viruses? This of course will lead to the (likely false) inferral that they are all novel... There is a huge database of viruses/phages derived from metagenomics (and were thus not cultivated) and the viral contigs should be compared to those? The authors then do compare them to “selected published environmental metagenomes”(384) but how and why were these selected?

Response #14: We agree that the viruses should be compared to both cultivable and uncultivable viral genomes in the databases when evaluating their novelty. In the revised version, we removed the analyses of comparing to the “*selected* published environmental metagenomes” (384). Instead, we included the viruses from both NCBI RefSeq and IMG/VR databases for taxonomic assignments. The later database contained uncultivated viral genomes from metagenomics. As a result, 139 vOTUs (instead of 3 in the original analyses that only included cultivable viruses for comparison) were assigned a taxonomy. We have modified the main text to reflect these changes:

Lines 221-226: “Taxonomic analyses, by comparing VLS viruses to viral genomes in both the NCBI RefSeq and IMG/VR databases (see Methods), revealed that most of the VLS long vOTUs (911 of 1,050) could not be taxonomically classified, indicating a high degree of novelty among VLS viruses. The remaining 139 vOTUs were assigned to Caudoviricetes, Faserviricetes, and Megaviricetes (Fig. 3A; Supplementary Fig. S7 & Table S9).”

Lines 486-491: “Because viruses lack any single, universally shared gene, we established taxonomy using gene-sharing network analysis from viral sequences ≥ 10 kb in length using vConTACT v2²³. Briefly, this analysis compared the 1,050 “long” VLS vOTUs and the 911

public datasets-originated viruses that contained MM AMGs to viral genomes in the National Center for Biotechnology Information (NCBI) RefSeq database (release v201) and the IMG/VR v4 database, and generated viral clusters approximately equivalent to known viral genera^{11,23,73}.”

In addition, Fig. 3A was modified to accommodate the changes in taxonomy assignments, and provided below for review convenience:

(A) Taxonomic assignments

- L:254: how do the viruses alter the ecophysiology of the sediments? That seems like a really huge claim with no evidence to support this

Response #15: We apologize for this overstated claim, as our data did not sufficiently support it. This sentence has been modified as (Lines 325-329):

“While no AMGs that can directly participate in MMP were detected from these methane-rich lake sediments, these results indicate that VLS viruses encode functional AMGs that likely alter microbial metabolisms in the Varana Lake sediments, including an AMG that have an indirect influence on MM through manipulating a putative methanogen’s iron metabolism.”

- L259-275: This reads like a summary or even abstract and not like a discussion.

Response #16: We agree and have shortened this paragraph to ¼ length in the revised version. Now it reads as (Lines 332-354):

“After carbon dioxide, methane is the second largest contributor to warming, accounting for approximately 20% of greenhouse gas-driven warming^{3,56,57}. While it is widely accepted that bacteria and archaea are major players in the global methane cycling, little was known about how viruses might impact MM. This study identified 24 virus-encoded MM AMGs in 911 viral contigs by analyzing 982 published metagenomes from environments where microbial MM is known to occur, and where microbial genes involved in MMP were detected. We found that the extent to which viruses use MM AMGs to modulate host MMP may vary depending on the ecological properties of the habitat in which they dwell. This finding is consistent with previous reports of the habitat-specific association of AMGs in the environments^{13,59}.”

- L284-287: where does this hypothesis come from and how does the data presented in this paper connect to it?

Response #17: We apologize for this mistake. We realized that this is an overspeculation and our data presented cannot mechanistically support this hypothesis. In the revised manuscript, we removed this hypothesis and added the suggestion that future studies are necessary to test the presence pattern of MM AMGs and its mechanism in the environments (Lines 396-399):

“Future studies are necessary to experimentally validate ... and to further test the presence pattern of MM AMGs and its mechanism as more host-associated and environmental metagenomes become available.”

- Figure 1A: The representation is extremely hard to follow. It also suggests that *pmoC* is in one cell (virus ?) with the other enzymes? Gene names are to be written in italics. This Figure should be revised.

Response #18: We apologize for the confusion, and thanks for your comments to help us improve the figure. The *pmoC* should not be visualized in one cell with the other enzymes. We have modified this plot by removing the cell cartoon, italicizing the gene names, and reorganizing the representation of MMP.

For review convenience, we provide the revised plot and the figure legend below:

(A) MM key steps with the seven exclusive MM AMGs' participation

Figure 1. Characterization of exclusive MM AMGs. (A) Schematic for viral participations in key MMP steps via encoding seven AMGs that exclusively participate in MMP. Viruses encoded seven AMGs (*fwdF*, *fae*, *frhB*, *cofE*, *cofF*, *mtrA* and *pmoC*; as colored in purple text) to impact the key steps in both methane production and oxidation. The methanogenesis pathway from CO₂ to methane is indicated by orange arrows. More information for 17 additional AMGs that could potentially participate in both MMP and other types of

metabolism pathways is provided in Supplementary Fig. S1, Fig. S2, & Table S5. **(B)** Genome maps ... MMP, methane metabolism pathway.

- Figure 1C: Is it necessary to show this tree in such a collapsed way? It seems to provide little information. As a sidenote: protein names should be capitalized.

Response #19: Thanks for your comments. We agree that this tree should not be visualized in such a collapsed way. It has been revised by removing some collapses and modifying the protein name “MtrA” as capitalized.

For review convenience, the updated figure is provided below:

(C) ML tree of MtrAs

- Figure 2: Is it necessary to show this as a main figure? I am more confused by looking at it than having read the section in the text, unfortunately.

Response #20: We apologize for the confusion. We prefer to keep it as a main figure, as these Venn plots visualize the *variation of MM AMGs's presence/absence by habitat*, with an emphasis on the finding that rumen, the host-associated habitat, contained the majority of virus-encoded MM AMGs by comparing to the environmental habitats. We choose Venn plots as they can easily visualize both the total number of AMGs in each environment (this is described in the text) and the number of AMGs shared between/among environments (not described in the text).

To address a comment on “*There are lots of pie charts and circles in several figures*” in Response #21, we have tried using UpSet plots, instead of the Venn plots that contained lots of circles, to visualize the AMG numbers:

(A) All 24 MM AMGs

(B) Ten MM AMGs that can participate in the pathway of methanogenesis from CO₂ or acetate

However, we found that the UpSet plots did not convey the findings better than Venn plots. Thus, we decided to use the Venn plots, but not the UpSet plots. In the revised version, we have modified the Venn plots and the figure legend to clearly convey the habitat-associated AMG distribution pattern as mentioned above.

For review convenience, we have included the revised figure and its legend below:

(A) All 24 MM AMGs

(B) Ten MM AMGs that can participate in the pathway of methanogenesis from CO₂ or acetate

Figure 2. Venn diagrams visualizing the number of MM AMGs from different habitats. (A) All the 24 MM AMGs. We identified 24 distinct MM AMGs from six habitats: rumen (16 AMGs), marine water (7), marine sediment (5), lake water (3), lake sediment (1), and hot spring sediment (2). Seven of these genes were identified in 2–4 habitats, and the remaining 17 were found exclusively in one of these habitats. **(B)** Ten AMGs involved in methanogenesis pathway. Of the 24 MM AMGs, 10 genes (i.e., *mtrA*, *fwdF*, *cofE*, *cofF*, *frhB*, *ackA*, *pta*, *cooS*, *glyA*, and *fae*) can directly participate in or synthesize an intermediate for the pathway of methanogenesis from CO₂ or acetate (Supplementary Fig. S1). Nine of these 10 AMGs were found in rumen, while only one to three were found from other detectable environmental habitats including marine water, marine sediment, lake water, and lake sediment. Three of these genes were identified in 2–4 habitats, and the remaining six and one were found exclusively in rumen and marine sediment, respectively.

- The Figures generally look a bit unpolished... e.g., Figure 3 (A) and (B) are outside the boxes, where (C) and (D) are inside. There are lots of pie charts and circles in several figures.. Figure 4A why does the green arrow restate in green font what is written in the left pie-chart? Were all 2,167 vOTUs assigned to either bacteria or archaea? This seems surprising that taxonomic classification/linkage is possible for every vOTU found..

Response #21: Thank you for your valuable comments to help us improve the figures. We have made significant revisions to all four main figures, including the modifications you highlighted in the comments. Among the revisions, while we removed one of the pie chart plots (the original Fig. 3B), we have retained the other pie charts and Venn plots (i.e., circles), as we consider them essential and appropriate for conveying key findings. We believe that the new figures represent a considerable improvement.

For the comment “Were all 2,167 vOTUs assigned to either bacteria or archaea? This seems surprising that taxonomic classification/linkage is possible for every vOTU found”, we apologize

for the confusion. We identified a total of 3,146 vOTUs, and 2,167 of them (not all of them) could be linked to their hosts. They were all predicted to infect either bacteria or archaea, as this study only identified bacterial and archaeal viruses and used VLS bacterial/archaeal genomes as the host database for host prediction. To clarify this, we modified the figure (i.e., Fig. 4A, see below), the main text (Lines 89-92 and 270-274), and figure legend (Lines 916-918):

Lines 88-92: “*We then generated an additional 11 metagenomes using ... sample bacterial/archaeal viruses and investigate their potential impacts on MM during infection.*”

Lines 269-273: “*Using the 99 VLS bacterial/archaeal MAGs as the host database (Supplementary Table S10; See Methods), we could link 2,167 of the 3,146 vOTUs (68.9%) to microbial hosts belonging to 17 different phyla (Fig. 4A; Supplementary Table S12).*”

Lines 946-948: “*Of the 3,146 vOTUs, 2,167 were linked to putative hosts, with 1,110 and 1,057 vOTUs infecting bacteria and archaea, respectively.*”

For review convenience, the updated version of Fig. 1, Fig. 3, and Fig. 4 are included below (Fig. 2 was provided in Response #20):

(A) MM key steps with the seven exclusive MM AMG's participation

(B) Genome maps of three viruses containing the AMG *mtrA*

(C) ML tree of MtrAs

Figure 1. Characterization of exclusive MM AMG's. (A) Schematic for viral participations in key MMP steps via encoding seven AMG's that exclusively participate in MMP. Viruses encoded seven AMG's (*fwdF*, *fae*, *frhB*, *cofE*, *cofF*, *mtrA* and *pmoC*; as colored in purple text) to impact the key steps in both methane production and oxidation. The methanogenesis pathway from CO₂ to methane is indicated by orange arrows. More information for 17 additional AMG's that could potentially participate in both MMP and other types of metabolism pathways is provided in Supplementary Fig. S1, Fig. S2, & Table S5. (B) Genome maps of three

viral contigs carrying the AMG *mtrA* gene. The three viral contigs belonged to the same viral population (with 97.4–97.8% genomic identities among each other) and carried an identical *mtrA* gene. CheckV was used to assess host-virus boundaries and remove potential host fractions on the viral contig. Genes were marked by five colors to illustrate AMGs (purple), phage genes (orange), phage hallmark genes (blue), potential cellular genes (green), and hypothetical protein genes (grey). (C) Phylogenetic tree of the viral and microbial *mtrA* genes. The tree was inferred using maximum likelihood method with protein sequences. Parametric bootstrap values (expressed as percentages of 1,000 replications) are shown at branching points. The scale bar indicates a distance of 0.1 substitutions per position in the alignment. The viral and microbial *MtrA* sequences are indicated in red and black, respectively. The numbers in parentheses indicate the number of protein sequences assigned to each group. The full phylogenetic tree (without collapsed groups) is provided in Supplementary Fig. S5A. The genomic maps and phylogenetic trees for the other six exclusive MM AMGs (*pmoC*, *fwdF*, *fae*, *cofE*, *cofF*, and *frhB*) are provided in Supplementary Fig. S3 and Fig. S5B–G. MM, methane metabolism; MMP, methane metabolism pathway.

(A) Taxonomic assignments

(B) vOTU number: Muddy vs. Sandy

(C) Community distribution

Figure 3. Viral communities of Vrana Lake sediments (VLS). (A) Taxonomic assignments of VLS vOTUs. The left chart shows the fraction of “long” vOTUs (length ≥ 10 kb) among all VLS vOTUs ($n = 3,146$). The right chart shows the taxonomy of VLS “long” vOTUs, when compared to viral genomes in the NCBI RefSeq and IMG/VR databases. Further details of taxonomic results are listed in Supplementary Table S9. (B) Shared and unique vOTUs between the two sediment sites (Muddy vs. Sandy sites) in Vrana Lake as shown in Supplementary Fig. S6. Only 4.2% of the 3,146 vOTUs were presented in both sites, while the remaining

95.8% were unique in either site. (C) PCoA plot of VLS samples based on the relative abundances of vOTUs. Samples are marked by the two sediment sites (Muddy and Sandy sites in green and orange, respectively) and the two metagenome types (bulk metagenome and virome as triangles and circles, respectively). PERMANOVA (permutations = 999) shows a statistically significant difference in viral communities between both sampling sites and metagenome types. The p values <0.05 are indicated in red.

(A) Predicted hosts

(B) Host and virus relative abundances for the two most abundant phyla

(C) Genome map of the vOTU “S225_M_175_17980bp” encoding the AMG *bfr*

(D) ML tree of Bfrs

Figure 4. VLS virus-host interactions. (A) Predicted phylum-level hosts of VLS viruses. Of the 3,146 vOTUs, 2,167 were linked to putative hosts, with 1,110 and 1,057 vOTUs infecting bacteria and archaea, respectively. The number of vOTUs putatively infecting each phylum is indicated in parentheses after the phylum name. (B) Relative abundances of the two most abundant VLS microbial phyla and their predicted

viruses. The two phyla are Thermoproteota (*left* panel of box plots) and Chloroflexi (*right* panel of box plots). Relative abundances of microbes and vOTUs were obtained based on their coverages generated by read mapping to MAG populations and vOTUs. Box-plot elements: center line, median; x symbol, mean; box range, upper and lower quartiles; whiskers, data range. (C) Genome map of the vOTU S225_M_175_17980bp encoding the AMG *bfr*. The genome content was characterized by the same methods described in Fig. 1B. (D) Phylogenetic tree of the viral and microbial *bfr* genes. The tree was inferred using maximum likelihood method with protein sequences (see Methods). Parametric bootstrap values (expressed as percentages of 1,000 replications) ≥ 40 are shown at branching points. The scale bar indicates a distance of 0.2 substitutions per position in the alignment. The viral *Bfr* sequence is indicated in red and other sequences are indicated in black. The full phylogenetic tree (without collapsed groups) is provided in Supplementary Fig. S11. VLS, Vrana Lake sediment.

Minor comments:

- L28: “methanogenic sediments” seems very colloquial, maybe rephrase to “sediments where methane emissions/consumption has been detected”(or whatever is applicable)

Response #22: The text has been rephrased based on your suggestion (Lines 88-90):

“We then generated an additional 11 metagenomes using lake sediments, in which methane emission has been detected, from Croatia’s largest freshwater lake ...”

- L90: resulting in the discovery..

Response #23: It was modified as “*resulting in the discovery*” in the revised main text (Line 115).

- L93L if only one protein sequence was investigated, how can the “conserved domains” be investigated ?

Response #24: We apologize for the confusion. The conserved domains were analyzed by comparing to the sequences in the domain database. To clarify, we modified the method sentence in the main text as (Lines 548-550):

“In addition, the AMGs of the selected viral contigs were further used as examples for analyzing conserved domains by comparing to the domains in the public Conserved Domain Database (CDD) via NCBI CD-Search⁸⁴ and ...”

- L104: on only one viral contig (not “from”)

Response #25: Thank you for identifying this mistake. We changed “from viral contigs” to “on viral contigs” through the text:

Lines 143-144: *“Among these seven AMGs, *fwdF* and *fae* were each detected on only one viral contig, while the others were identified on 3 to 25 viral contigs ...”.*

Lines 538-539: *“a discovery of 24 distinct AMGs (on a total of 911 viral contigs containing ≥ 1 MM AMG) ...”.*

Line 546: “the AMG was presented on more than one contig”.

Line 562: “if an AMG was identified on more than two viral contigs ...”.

- L106: it should be stated that pmoC is part of the aerobic methane oxidation pathway of bacteria

Response #26: This was stated in the revised text (Lines 146-152):

“Functionally, the pmoC gene participates in the aerobic methane oxidation pathway of bacterial methanotrophs²⁷, while the other six genes mtrA, fwdF, fae, cofE, cofF, and frhB are involved in the pathways of methanogenesis and/or anaerobic oxidation of methane (AOM)²⁸ (Fig. 1A & Supplementary Fig. S1). The pmoC (methane monooxygenase subunit C) gene encodes a subunit of the particulate methane monooxygenase (pMMO) that catalyzes the aerobic oxidation of methane to methanol in bacteria (Fig. 1A & Supplementary Fig. S1)²⁷.”

- L107: how do the authors know that these genes are not involved in AOM? Unless there is evidence for this (e.g., the metagenomic samples these viral contigs originate from, only comprise methanogens, and not ANME). If this was not investigated, it should be added “pathway of methanogenesis or reverse methanogenesis”

Response #27: Thank you for pointing out this mistake. There is no evidence to support that these genes are not involved in AOM. We have revised text as (Lines 147-149):

“while the other six genes mtrA, fwdF, fae, cofE, cofF, and frhB are involved in the pathways of methanogenesis and/or anaerobic oxidation of methane (AOM)²⁸ (Fig. 1A & Supplementary Fig. S1).”

- L168: how were the vOTUs established?

Response #28: Viral contigs were clustered into vOTUs if they shared $\geq 95\%$ nucleotide identity across 80% of their length. We have modified this sentence to include the methods (Lines 216-219):

“We recovered 3,260 viral contigs from these metagenomes. These contigs were clustered into vOTUs if they shared $\geq 95\%$ nucleotide identity across 80% of their lengths³⁸, resulting in 3,146 vOTUs (≥ 5 kb), including 1,050 “long” (≥ 10 kb) vOTUs (Supplementary Table S8).”

The method was also described in the Methods section of the original main text (new Lines 467-469):

“The remaining contigs were clustered into vOTUs (~species-level taxonomic unit) if they shared $\geq 95\%$ nucleotide identity across 80% of their lengths as described previously³⁸”

- L 211: how were vOTUs linked to microbial hosts? This should be stated in brief in the results.

Response #29: We have modified this sentence to briefly describe how the vOTUs were linked to microbial hosts (Lines 267-273):

“To explore the potential viral impacts on VLS ecosystems, we investigated virus-host linkages as reported previously (e.g., in soil and seawater^{11,39}), via the iVirus tool VirMatcher⁴⁰ that aggregates four different methods for host predictions (see Methods). Using the 99 VLS bacterial/archaeal MAGs as the host database (Supplementary Table S10; See Methods), we could link 2,167 of the 3,146 vOTUs (68.9%) to microbial hosts belonging to 17 different phyla (Fig. 4A; Supplementary Table S12).”

- L221: I don't think it is accurate to write “many” Thermoproteota members are methanogens. Also not sure why “archaeal” is required at this point, because all methanogens (we know so far!) are archaeal (this would fit in the beginning, but not here)

Response #30: We agree and have modified this sentence as (Lines 281-283):

“Some MAGs of Thermoproteota contained genes encoding for key steps of MM⁴¹⁻⁴⁴ ...”

We also modified the text in the introduction to state that microbial methane production is mediated solely by archaea (Lines 62-64):

“Methane cycling is largely driven by microbes, with microbial methanogenesis (all mediated by archaea) producing ~69% of the total methane released to the atmosphere⁶.”

Per the other reviewer's suggestion, we also provided gene annotations for all the *Thermoproteota* MAGs (n = 23) we have recovered from VLS metagenomes and highlighted their genes involved in MMP. Each of the *Thermoproteota* MAGs contained 24–102 (average 65) genes involved in MMP. Please see Response #9 for further details.

- L224: had an important

Response #31: The phrase “had important” was modified to “had an important” in the revised text (Line 289).

- L223-5: This seems pretty disconnected. What is the evidence for this or why bring this up here?

Response #32: Thank you for bringing this to our attention. We agree and have removed the text “Based on the classical Kill-the-Winner model” from this sentence. The modified text reads as (Lines 287-289):

“Overall, these findings showed that the VLS viruses infected dominant microbial phyla, including ones involved in MM, and thus likely had an important impact on the sediment ecosystems.”

- L242: “conserved functional domains of their corresponding enzymes” What does this mean?

Response #33: We apologize for the confusion. Our intention was to convey that all the AMG's possessed conserved functional domains. We have modified the text as (Lines 319-320):

“All of them had the conserved functional domains (Supplementary Table S15)”

- L:251: Why was this pN/pS analysis done?

Response #34: We calculated the pN/pS values for these AMG's, because the published standards methodology for studying virus-encoded AMG's (Pratama *et al.*, 2021. PeerJ) recommends investigating gene's selection pressure (e.g., pN/pS value) which indirectly indicates AMG's evolution and functionality information.

Pratama AA, Bolduc B, Zayed AA, Zhong ZP, Guo J, Vik DR, Gazitua MC, Wainaina JM, Roux S & Sullivan MB (2021) Expanding standards in viromics: in silico evaluation of dsDNA viral genome identification, classification, and auxiliary metabolic gene curation. *PeerJ* 9: e11447. DOI: <https://doi.org/10.7717/peerj.11447>.

- L282: “significant” may not be the best term here

Response #35: We have removed these sentences in the revised version based on another comment (see Response #16).

- L293: CO is also a substrate for methanogens

Response #36: Thanks for identifying this omission. We have modified the sentence to clarify that CO is also a substrate for methanogens (Lines 365-370):

“For example, the carbon monoxide (CO) dehydrogenase gene (cooS; identified from a rumen metagenome; Supplementary Table S5) catalyzes the oxidation of CO to CO₂⁶⁰, which is the substrate of a methanogenesis pathway from CO₂ in ruminants⁶¹. In addition, the oxidation of CO to CO₂ in itself could be a step of methanogenesis using CO as the substrate⁶² and an energy generating metabolic reaction⁶³.”

- L:294: It is a bit odd to have (n=17) in the discussion section as such specific values usually belong to the results

Response #37: The phrase “(n=17)” was removed from the sentence, now it reads as (Lines 371-373):

“Thus, while many of the identified AMG's have the potential to participate in MM, without further verification, their actual functions remain hypothetical.”

- L297ff: it should be stated that mcr genes were recently found in “Borgs” – archaeal ECEs that are neither typical viruses nor plasmids

Response #38: We agree and have added a statement to the text (Lines 376-379):

“Interestingly, while not yet found in viral genomes, the mcr genes were recently discovered in a novel group of extrachromosomal elements called “Borgs”, that are associated with ANME in the genus Methanoperedens¹⁵”

REVIEWERS' COMMENTS

Reviewer #1 (Remarks to the Author):

This manuscript has improved substantially during revisions, and the authors have responded to most of the points raised prior. I have some comments that I still think could improve the MS further:

1. There should be a figure describing the predicted iPhoP hosts of the viruses with MM AMGs. This could be added to Figure 2.
2. There is no evidence here that the viruses actually do modulate microbial methane production. I would encourage the authors to reframe their title in terms of their primary finding, which is that viral genomes encode MM AMGs.
3. Throughout the manuscript "KEGG annotations" are referred to without properly specifying the methods used to annotate KEGG orthologs. Were HMMs used? KOFams or METABOLIC? What cutoffs were used? Please be specific and cite, as all annotations are not equal. Apologies if I missed it.
4. The authors still need to put their own data in the context of their overall analysis in the discussion. On average in their global analysis, what % of metagenomics from these environments had at least one viral MM AMG? With that number in mind, is it unusual or expected that they did not recover one from their own lake samples?

Minor points:

1. iPhoP is not mentioned in the methods - please add it.
2. The assembler used should be stated directly in the methods.
3. In the methods there is a dn/ds analysis described that I do not think appears in the text.
4. I think it is poor communication to state that Methane is a "more potent" greenhouse gas than CO₂, as this can be misleading. I think the more appropriate statistic is the cited "After carbon dioxide, methane is the second largest contributor to warming, accounting for approximately 20% of greenhouse gas-driven warming" in the discussion, and this fact should be used in the intro instead of the "potent" comparison.
5. Please state how deeply samples were sequenced (Gbp) in the methods.

Reviewer #2 (Remarks to the Author):

Upon careful review of the revised manuscript by Zhong et al., I am pleased to note that the authors have adequately addressed my previous concerns and suggestions. The revisions have significantly improved the clarity and quality of the work. Consequently, I can now recommend this manuscript for publication. I believe it makes a valuable contribution to the field and will be of interest to the readership of Nature Communications.

REVIEWERS' COMMENTS

Reviewer #1 (Remarks to the Author):

This manuscript has improved substantially during revisions, and the authors have responded to most of the points raised prior. I have some comments that I still think could improve the MS further:

Response #1: We appreciate your kind words in assessing our revisions and thank you for the additional comments, which we have implemented into this revision as described below. Please note that all the line numbers cited in our responses are based on the revised manuscript with tracked changes.

1. There should be a figure describing the predicted iPhoP hosts of the viruses with MM AMG. This could be added to Figure 2.

Response #2: We agree and have modified Figure 2 to include the information on host prediction as Fig. 2A. The original Fig. 2A and 2B were changed to Fig. 2B and 2C, respectively.

For review convenience, we provide Fig. 2 and the figure legend below:

(A) Predicted hosts of viuses containing MM AMGs

(B) Habitat association of all 24 MM AMGs

(C) Habitat association of the 10 MM AMGs involved in the pathway of methanogenesis from CO₂ or acetate

Figure 2. Predicted hosts of viruses encoding MM AMGs and habitat association of MM AMGs. (A) Phylum-level host predictions of 257 viruses that encoded MM AMGs. Of the 911 viral contigs encoding MM AMGs, 257 were successfully linked to hosts that belonged to four archaeal and 10 bacterial phyla. Additional information about the predicted hosts is provided in Supplementary Table S4. (B–C) Habitat association of all the 24 MM AMGs (B) and the 10 MM AMGs involved in methanogenesis pathway (C). We identified 24 distinct MM AMGs from six habitats: rumen (16 AMGs), marine water (7), marine sediment (5), lake water (3), lake sediment (1), and hot spring sediment (2). Seven of these genes were identified in 2–4 habitats, and the remaining 17 were found exclusively in one of these habitats. Of the 24 MM AMGs, 10 genes (i.e., *mtrA*, *fwdF*, *cofE*, *cofF*, *frhB*, *ackA*, *pta*, *cooS*, *glyA*, and *fae*) can directly participate in or synthesize an intermediate for the pathway of methanogenesis from CO₂ or acetate (Supplementary Fig. S1). Nine of these 10 AMGs were found in rumen, while only one to three were found from other detectable environmental habitats including marine water, marine sediment, lake water, and lake sediment. Three of these genes were identified in 2–4 habitats, and the remaining six and one were found exclusively in rumen and marine sediment, respectively.

2. There is no evidence here that the viruses actually do modulate microbial methane production. I would encourage the authors to reframe their title in terms of their primary finding, which is that viral genomes encode MM AMGs.

Response #3: We have modified the title from “*Viral modulation of microbial methane production and oxidation varies by habitat*” to “*Viral potential to modulate microbial methane metabolism varies by habitat*”.

3. Throughout the manuscript "KEGG annotations" are referred to without properly specifying the methods used to annotate KEGG orthologs. Were HMMs used? KOFams or METABOLIC? What cutoffs were used? Please be specific and cite, as all annotations are not equal. Apologies if I missed it.

Response #4: We apologize for this omission. We used published tools to annotate genes against the KEGG database. The viral and microbial contigs were annotated by the tools VIBRIANT and DRAM, respectively (Kieft et al., 2020 *Microbiome*; Shaffer et al., 2020 *Nucleic Acids Res*), using their default parameters. We added the below sentences to the *Methods* section to clarify the gene annotation methods:

Lines 484–486: “*Specifically, once viral contigs were recovered from metagenomes, they were processed with VIBRIANT to obtain gene functional annotations against the KEGG and PFAM databases and identify putative AMGs by the default parameters*²².”

Lines 514–517: “*The assembled contigs, excluding viral contigs, were annotated by DRAM against the KEGG and PFAM databases by the default parameters and further used for recovering microbial MM genes based on their KEGG and PFAM annotations (with consistent annotation in the two databases) and the KEGG MM pathway modules*¹⁸.”

Kieft K, Zhou Z & Anantharaman K (2020) VIBRIANT: automated recovery, annotation and curation of microbial viruses, and evaluation of viral community function from genomic sequences. *Microbiome* **8**: 90.

Shaffer M, Borton MA, McGivern BB, Zayed AA, La Rosa SL, Solden LM, Liu P, Narrowe AB, Rodriguez-Ramos J, Bolduc B, Gazitua MC, Daly RA, Smith GJ, Vik DR, Pope PB, Sullivan MB, Roux S, & Wrighton

KC (2020) DRAM for distilling microbial metabolism to automate the curation of microbiome function. *Nucleic Acids Res* **48**: 8883-8900.

4. The authors still need to put their own data in the context of their overall analysis in the discussion. On average in their global analysis, what % of metagenomics from these environments had at least one viral MM AMG? With that number in mind, is it unusual or expected that they did not recover one from their own lake samples?

Response #5: On average, 32.2% of the metagenomes in the global analysis had at least one viral MM AMGs, whereas only 1.7% of the lake-sediment metagenomes contained at least one viral MM AMGs. Thus, the finding that we did not recover one MM AMG from the 11 Vrana Lake sediment metagenomes (our own data) would agree with the analyses of the public data. The below sentences, including one in the *Discussion* section, have been modified to contextualize the above arguments:

Lines 118-119: *“These 911 viral contigs originated from ~32% (316 of 982) of the here analyzed metagenomes (Supplementary Table S3).”*

Lines 283-288: *“Interestingly, none of these AMGs were predicted to be directly involved in the MMP, which would agree with the inference of our analysis of publicly available metagenomes that the extent to which viruses modulate hosts’ MM may vary by habitats, and that MM AMGs seem to be less common in environmental habitats including lake sediments (<2% of the publicly available lake-sediment metagenomes had ≥ 1 MM AMG).”*

Lines 318-324 (in the *Discussion* section): *“We found that the extent to which viruses use MM AMGs to modulate host MMP may vary depending on the ecological properties of the habitat in which they dwell. Specifically in lake sediments, less than 2% of the publicly available metagenomes contained ≥ 1 MM AMG and no MM AMG was identified from the 11 metagenomes of Vrana Lake sediments, in which methane emission has been detected. This finding is consistent with previous reports of the habitat-specific association of AMGs in the environments^{15,57}.”*

Minor points:

1. iPhoP is not mentioned in the methods - please add it.

Response #6: We apologize for this mistake. We have added it to the *Methods* section of the revised manuscript (Lines 501-504):

“Hosts of the 911 viral contigs containing ≥ 1 MM AMG were predicted by iPhoP²⁶ (confidence score $\geq 90\%$), resulting successfully virus-host linkages for 257 viral contigs (Supplementary Table S4).”

2. The assembler used should be stated directly in the methods.

Response #7: We added the assembler information to the *Methods* section in the revised version (Lines 415-420).

“Sequencing reads were filtered for quality ... Then the metagenomic sequence data was assembled to contigs by metaSPAdes⁶⁷, using a previously established pipeline for

assembling pre-amplified metagenomes (parameters: read deduplication + read error correction + --sc + -k 21,33,55,77,99,127)⁶⁸.”

3. In the methods there is a dn/ds analysis described that I do not think appears in the text.

Response #8: Thanks for pointing out this omission. The *dS/dS* result was described in the Supplementary Information in the last version. In the revised version, we also included this result in the main text (Lines 295-297):

“Evolutionary pressure assessments within species and across lineages found that this virus-encoded Bfr was likely functional and under purification selection ($pN/pS = 0$; average $dN/dS = 0.114$; Supplementary Table S15 & Table S16).”

4. I think it is poor communication to state that Methane is a "more potent" greenhouse gas than CO₂, as this can be misleading. I think the more appropriate statistic is the cited "After carbon dioxide, methane is the second largest contributor to warming, accounting for approximately 20% of greenhouse gas-driven warming" in the discussion, and this fact should be used in the intro instead of the "potent" comparison.

Response #9: We have modified the sentence that has the phrase “*more potent*” in the introduction based on your suggestion. Now it reads as (Line 62-64):

“Methane (CH₄) is ranked second after carbon dioxide (CO₂) in terms of the overall contribution to atmospheric warming and accounts for ~20% of the greenhouse gas-driven warming³⁻⁵.”

5. Please state how deeply samples were sequenced (Gbp) in the methods.

Response #10: The sequencing depth of each sample was provided in Supplementary Table S8 in the last version, and now we also included this information in the *Methods* section (Lines 415-417):

“Sequencing reads were filtered for quality ... generating a total of 9.5×10^{10} bases of sequencing data (range $0.3-1.5 \times 10^{10}$ bases, average 8.6×10^9 bases per library; Supplementary Table S8).”

Reviewer #2 (Remarks to the Author):

Upon careful review of the revised manuscript by Zhong et al., I am pleased to note that the authors have adequately addressed my previous concerns and suggestions. The revisions have significantly improved the clarity and quality of the work. Consequently, I can now recommend this manuscript for publication. I believe it makes a valuable contribution to the field and will be of interest to the readership of Nature Communications.

Response #11: We appreciate your kind evaluations of our revisions and thank you for recommending our work for publication at *Nature Communications*.